# Therapeutic high affinity T cell receptor targeting a KRAS^G12D cancer neoantigen

Andrew Poole [1], Vijaykumar Karuppiah [1], Annabelle Hartt [2], Jaafar N. Haidar[3], Sylvie Moureau[1], Tomasz Dobrzycki [1], Conor Hayes[1], Christopher Rowley[1], Jorge Dias[1], Stephen Harper[1], Keir Barnbrook [1], Miriam Hock[1], Charlotte Coles[1], Wei Yang[3], Milos Aleksic[1], Aimee Bence Lin[3], Ross Robinson[1], Joe D. Dukes[1], Nathaniel Liddy[1], Marc Van der Kamp [2], Gregory D. Plowman [3], Annelise Vuidepot[1], David K. Cole[1], Andrew D. Whale [1] ✉ & Chandramouli Chillakuri [1] ✉

Neoantigens derived from somatic mutations are specific to cancer cells and are ideal targets for cancer immunotherapy. *KRAS* is the most frequently mutated oncogene and drives the pathogenesis of several cancers. Here we show the identification and development of an affinity-enhanced T cell receptor (TCR) that recognizes a peptide derived from the most common KRAS mutant, KRAS^G12D, presented in the context of HLA-A*11:01. The affinity of the engineered TCR is increased by over one million-fold yet fully able to distinguish KRAS^G12D over KRAS^WT. While crystal structures reveal few discernible differences in TCR interactions with KRAS^WT versus KRAS^G12D, thermodynamic analysis and molecular dynamics simulations reveal that TCR specificity is driven by differences in indirect electrostatic interactions. The affinity enhanced TCR, fused to a humanized anti-CD3 scFv, enables selective killing of cancer cells expressing KRAS^G12D. Our work thus reveals a molecular mechanism that drives TCR selectivity and describes a soluble bispecific molecule with therapeutic potential against cancers harboring a common shared neoantigen.

Somatic mutations can create neoantigens that are unique to cancer cells and absent in normal tissues, making them attractive targets for immunotherapy[1–4]. In recent years, it has become evident that host immunity targets neoantigens to elicit endogenous T cell-mediated anti-tumor responses[5,6]. Although the vast majority of somatic mutations are random passenger mutations that are unique to an individual patient, a subset of mutations in oncogenic driver genes such as *KRAS*, *PIK3CA* or *ERBB2* and tumor suppressors including *TP53* and *PTEN* form a class of shared neoantigens common amongst patient cohorts[7,8].

*KRAS* is the most frequently mutated oncogene; mutations are found in ~14% of all cancers, with particularly high frequency in

pancreatic and colorectal cancers[9]. *KRAS* encodes a small GTPase, which cycles between an inactive GDP bound form and an active GTP bound form that propagates signal transduction pathways regulating cell survival, growth, and differentiation[10]. This core function of KRAS is compromised when somatic gain-of-function mutations inhibit the intrinsic GTPase activity leading to constitutive oncogenic signaling[11,12]. A hot spot for mutation in KRAS is glycine at position 12 of the protein, and the most frequent mutations at this residue are G12D followed by G12V and G12C[13].

Despite its well-established role in oncogenesis, KRAS has historically proved a challenging target for therapeutic intervention.

¹Immunocore Ltd., 92 Park Drive, Milton Park, Abingdon OX14 4RY, USA. ²School of Biochemistry, University of Bristol, Biomedical Sciences Building, University Walk, Bristol BS8 1TD, USA. ³Eli Lilly & Co, Lilly Corporate Center, Indianapolis, IN 46285, USA. ✉e-mail: andrew.whale@immunocore.com; chandramouli.chillakuri@immunocore.com

As an intracellular protein, KRAS is inaccessible to conventional monoclonal antibodies, while the lack of an amenable mutant-specific binding pocket has hindered attempts to develop selective small molecule antagonists[14]. Recently, AMG510 (sotorasib) and MRTX849 (adagrasib), inhibitors capable of binding irreversibly to KRAS[G12C], have demonstrated promising clinical anti-tumor activity[15–17]. However, these molecules target a less frequent mutation and so novel approaches are required to treat cancers driven by more frequent mutations, such as KRAS[G12D].

Unlike antibodies that, in the context of cancer recognition, primarily bind to intact cell surface expressed proteins, T cells can access mutated or dysregulated intracellular proteins via T cell receptor (TCR) recognition of neoantigen derived peptides presented by human leucocyte antigen (HLA) class I on the cancer cell surface. Thus, exploiting T cell mediated tumor immunity offers an alternative approach to target cancer specific neoantigens[18]. In support of this approach, a T cell clone specific for a KRAS[G12D] peptide presented in the context of HLA-C*08:02 has been successfully utilized in immunotherapy to suppress tumors in a patient[19]. In addition, a murine TCR isolated from humanized mice[20], and a bispecific T cell engaging TCR-mimetic antibody targeting KRAS[G12V] peptides in the context of HLA-A*11:01 (HLA-A*11) and HLA-A*03:01[3], respectively, have demonstrated T cell activation and killing of cancer cells preclinically. These pioneering studies have opened avenues to target KRAS mutant neoepitopes using peptide-HLA (pHLA) specific therapeutics[6]. However, the mechanisms underlying specificity to these neoantigens remains poorly understood.

Here, we isolate and affinity-enhance a human TCR specific to a KRAS[G12D] decamer peptide (VVVGADGVGK) presented in the context of HLA-A*11 (HLA-A*11-KRAS[G12D]). A combination of structural, biochemical, and computational approaches show that the TCR intricately achieves peptide selectivity from differences in electrostatic interactions, despite minimal changes in direct TCR-pHLA contacts, providing a detailed mechanistic understanding of the selectivity of this TCR to HLA-A*11-KRAS[G12D] over HLA-A*11-KRAS[WT]. Using this affinity enhanced TCR as the targeting arm, a bispecific T cell engaging ImmTAC (Immune mobilizing monoclonal TCR Against Cancer) molecule[21], mediates selective T cell targeting of HLA-A*11+ cancer cells naturally expressing KRAS[G12D]. This work highlights the exquisite sensitivity of the TCR:pHLA system and implies that soluble high affinity TCR bispecifics may have the potential to treat neoantigen driven cancers.

## Results

### Identification and characterisation of a KRAS[G12D] specific TCR
We identified a KRAS[G12D] specific human TCR (JDI TCR) from the peripheral blood mononuclear cells (PBMC) of an HLA-A*11+ healthy donor (JDI). T cells transduced with the JDI TCR, encoded by *TRAV19*01* and *TRBV6-2*01*, were co-cultured with peptide-pulsed HLA-A*11+ acute lymphoblastic leukemia SUP-B15 B cells. Concentration dependent IFNγ release was detected in co-cultures of JDI TCR-transduced cells and SUP-B15 cells pulsed with KRAS[G12D] decamer peptide (VVVGADGVGK), but not in response to KRAS[WT] peptide (VVVGAGGVGK), or unpulsed controls (Fig. 1a). Consistent with these data, the JDI TCR bound to HLA-A*11-KRAS[G12D] with an affinity ($K_D$) of 63 μM, with no measurable binding affinity to HLA-A*11-KRAS[WT] (Table 1).

To further assess JDI TCR specificity, we analyzed binding to several peptides from other RAS superfamily GTPase proteins with high amino acid sequence similarity, as well as pools of HLA-A*11 presented nonamer/decamer self-peptides from ubiquitously expressed genes (Supplementary Table S1). No measurable binding was detected to any of these pHLAs indicating a high level of selectivity towards KRAS[G12D] (Table 1). Alanine substitution at each position of the peptide indicated that peptide residue D6, along with G4, A5, and G9,

were most critical for JDI TCR binding (Fig. 1b), demonstrating the importance of the KRAS[G12D] mutation in mediating TCR recognition.

### Affinity-enhanced JDI TCRs retain the ability to distinguish KRAS[G12D] from KRAS[WT]
Strengthening TCR affinity enables effective targeting of antigens presented at low levels on the cell surface while longer $t_{1/2}$ may improve residence time of the TCR on antigen presenting cells to elicit persistent and effective T cell activation and target cell killing[22]. We affinity enhanced the HLA-A*11-KRAS[G12D] specific JDI TCR using NNK randomisation of complementarity determining regions (CDR)[23,24]. To ensure that discrimination between the neoepitope KRAS[G12D] and KRAS[WT] self-peptide was maintained or enhanced, affinity variant phage libraries were depleted for HLA-A*11-KRAS[WT] binders prior to positive selection on HLA-A*11-KRAS[G12D]. Binding to the ubiquitously expressed HLA-A*11-KRAS[WT] was monitored throughout the affinity maturation process and only the most specific mutants were used to combine and enhance the affinity in an iterative process. This procedure enabled incremental improvements in the JDI TCR affinity for HLA-A*11-KRAS[G12D] (Fig. 1c), whilst widening the affinity window ($K_D$ KRAS[WT]/$K_D$ KRAS[G12D] pHLA, Fig. 1d) from 98-fold for the JDIa9bwt TCR to >4000-fold for the affinity enhanced JDIa41b1 TCR and JDIa96b35 TCR compared to binding to HLA-A*11-KRAS[WT] (Fig. 1e, Table 2, Supplementary Fig. S1). Several affinity-enhanced JDI TCRs were generated with $K_D$ in the range of 50-100 pM and $t_{1/2} \geq 20$ hr representing ~ a one million-fold affinity enhancement over the parent JDI TCR with enhanced selectivity relative to the ubiquitously expressed HLA-A*11-KRAS[WT].

### The JDIa41b1 TCR adopts a virtually identical binding mode in complex with HLA-A*11-KRAS[G12D] and HLA-A*11-KRAS[WT]
To understand the molecular basis of TCR selectivity for HLA-A*11-KRAS[G12D], we solved the crystal structures of the JDIa41b1 TCR in complex with HLA-A*11-KRAS[G12D] ($K_D = 743 \pm 18$ pM) and HLA-A*11-KRAS[WT] ($K_D = 3.0$ μM) at 2.58 Å (JDIa41b1-HLA-A*11-KRAS[G12D] complex, PDB 7OW6) and 2.64 Å (JDIa41b1-HLA-A*11-KRAS[WT] complex, PDB 7OW5) resolution, respectively (Table 3). A superposition of the JDIa41b1-HLA-A*11-KRAS[WT] and JDIa41b1-HLA-A*11-KRAS[G12D] complexes revealed a virtually identical conformation (Root mean square deviation (RMSD) of TCR variable domains: 0.29 Å for Cα atoms and 0.80 Å for all atoms) (Fig. 2a), an identical crossing angle of 50° (Fig. 2b), with all CDRs, except CDR2β, directly contributing to the binding interface. Composite omit maps of the KRAS[WT] and KRAS[G12D] peptides and TCR CDR regions in the JDIa41b1-HLA-A*11-KRAS[WT] and JDIa41b1-HLA-A*11-KRAS[G12D] complexes, showed that modeled residues at the binding interface agree well with the observed experimental data. For a few CDR residues (R28α, Q57α), the side-chain densities were relatively weak (Supplementary Fig. S2), but the modeled side-chain structures do not differ substantially between the JDIa41b1-HLA-A*11-KRAS[WT] and JDIa41b1-HLA-A*11-KRAS[G12D] complexes. Only R28α contributes significantly to differences in binding affinity (based on the decomposition of binding energy calculated from MD simulations) and is discussed below. Analysis of the JDIa41b1-HLA-A*11-KRAS[WT] and JDIa41b1-HLA-A*11-KRAS[G12D] binding interfaces revealed that both complexes were very similar, with the JDIa41b1 TCR engaging mainly the C-terminus of the peptide through the CDR3β (Fig. 2c). Polar contacts to the peptide backbone were made through TCR residues Y33α (to peptide A5), N101β and H100β (to peptide G7) and G98β (to peptide G9).

TCR-HLA interactions were also identical in both complexes and were dominated by TCR α-chain contacts to the HLAα1 helix. Key JDIa41b1 TCR residues at the HLA interface included CDR1α residues R28α and D29α that formed salt bridges to HLAα1 residues E58 and R65, respectively, and CDR3α residues P97α, G99α, D100α

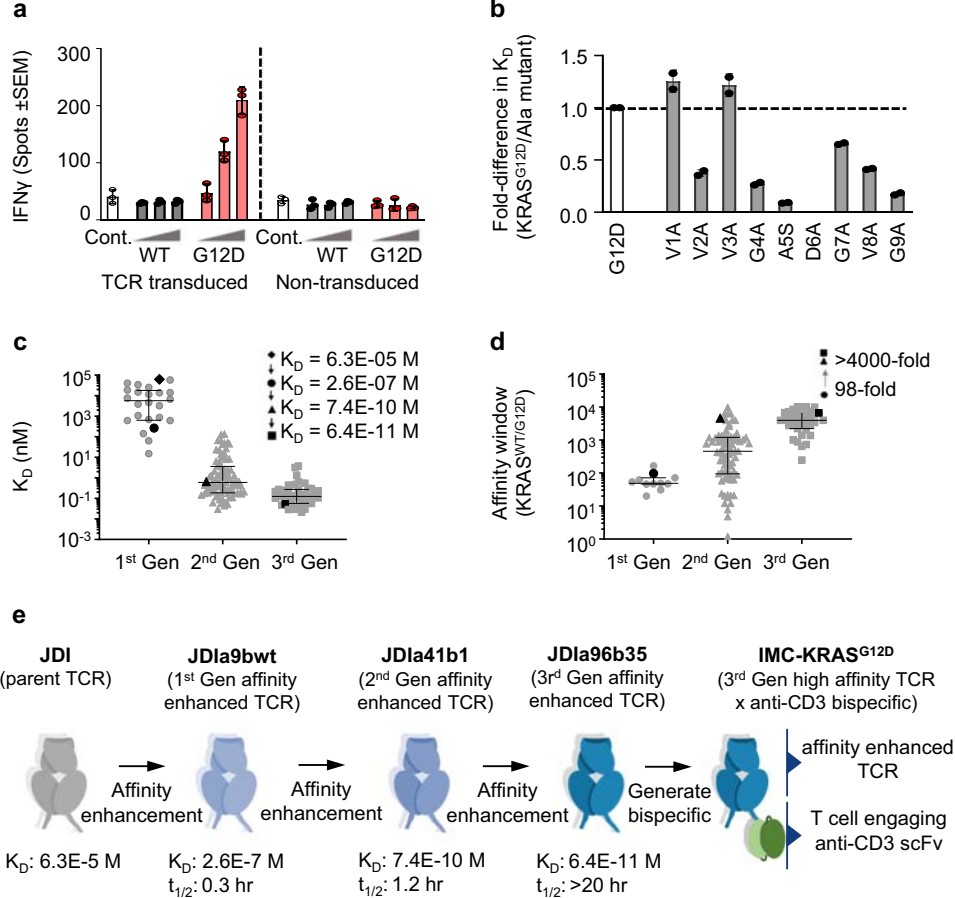

**Fig. 1 | Affinity-enhanced version of the JDI TCR retains the ability to distinguish KRAS^G12D^ from KRAS^WT^. a** T cells transduced with the JDI TCR were co-cultured with SUP-B15 cells (HLA-A*03:01 + /HLA-A*11 + ) incubated with 0.1, 1 or 10 μM KRAS^WT^ (WT) or KRAS^G12D^ (G12D) peptide. T cell activation was assessed using IFNγ capture ELISPOT assay. Unpulsed antigen presenting cells (Cont.) and untransduced T cells were used as controls. Mean data of three biological replicates ± SEM from one experiment is shown **b** binding of the JDI TCR to alanine substituted KRAS^G12D^ peptides. Mean $K_D$ values (n = 2) plotted as fold-change in $K_D$ compared to KRAS^G12D^ (G12D). Ratio is calculated as $K_D$ of JDI TCR binding to KRAS^G12D^/Alanine mutant. **c** Affinity ($K_D$) of mutants from three generations of affinity maturation. Error bars represent median with interquartile range in each generation (Gen; 1st Gen n = 24, 2nd Gen n = 78, 3rd Gen n = 41). Data shown in black symbols are for selected mutants shown in Fig. 1e and calculated from at least n = 2 experiments. **d** Difference in binding affinity of TCR mutants towards KRAS^WT^ and KRAS^G12D^ plotted as affinity window ($K_D$ KRAS^WT^/ $K_D$ KRAS^G12D^ pHLA). Error bars represent median with interquartile range in each generation (1st Gen n = 11, 2nd Gen n = 78, 3rd Gen n = 35). Data shown in black symbols are calculated from at least n = 2 experiments. **e** Schematic representation of the evolution of parent JDI TCR into a high affinity TCRxCD3-bispecific IMC-KRAS^G12D^. Source data are provided as a Source Data file.

and G101α that contacted HLAα1 residues R65, K68 and A69 (Supplementary Fig. S3b). Additional interactions were made through the TCR CDR3β residues P97β, G98β and H100β, that engaged HLAα1 helix residues A69, Q72 and T73, TCR CDR1β residue E30β that formed a salt bridge with HLA residue R75 (Supplementary Fig. S3a), TCR CDR1α residue T30α that bound to HLAα2 residues R163, and TCR CDR2α residues W53α and W54α that made hydrophobic contacts with HLAα2 residues Q155 and E154 (Supplementary Fig. S3c). A summary of the contacts for both complexes are listed in supplementary Table S2.

## KRAS^G12D^ peptide D6 side chain was buried in the HLA groove of the HLA-A*11-KRAS^G12D^ -TCR complex

Although there were no distinguishable differences in the interactions between the JDIa41b1 TCR and the KRAS^WT^/KRAS^G12D^ peptides, there were crucial differences between the HLA-A*11-KRAS^WT^/KRAS^G12D^ peptide interactions. D6 in the KRAS^G12D^ peptide was buried in the HLA groove and formed a salt bridge with R114 in the F-pocket of the HLA, which in turn was stabilized by HLA residue D116 (Fig. 2d). The D6 side chain also made a hydrogen bond to HLA residue Q70 (Fig. 2d). These interactions were not possible with the shorter side chain of G6 in the KRAS^WT^ peptide.

## KRAS^G12D^ peptide conformation does not depend on affinity of bound TCR

To ascertain if the JDI parent TCR also binds to HLA-A*11-KRAS^G12D^ in a similar fashion to the JDIa41b1 TCR, and to decipher the molecular determinants of affinity enhancement, we solved the crystal structure of JDI TCR-HLA-A*11-KRAS^G12D^ complex at 3.41 Å resolution (PDB 7PB2). The conformation of KRAS^G12D^ peptide was remarkably similar in the complex with weak affinity JDI TCR ($K_D$ 63 μM) and strong affinity JDIa41b1 TCR ($K_D$ 743 pM) suggesting that the conformation was not dependent on the affinity of TCR (Supplementary Fig. S4a, b). Superposition of the peptides gave RMSD values of 0.77 Å for all atoms and 0.23 Å for only Cα atoms. Minor conformational differences were observed in the side chain rotamers adopted by peptide residues V3 and V8 in the two complexes. V3 side chain was buried in the HLA groove and V8 side chain was partially exposed to the solvent (Supplementary Fig. S4c, d). JDIa41b1 TCR has 12 mutations covering all TCRα CDRs and CDR3β (Table 4). Of these mutations, only four changes (all in TCRα) were within 5 Å of the pHLA surface in the JDIa41b1-HLA-A*11-KRAS^G12D^ complex. Most notably, tryptophan residues introduced in CDR2α, slightly altered the loop conformation, and directly engaged with the peptide likely contributing to improved affinity. CDR2α mutation S53W, accommodated by concurrent CDR1α

**Table 1 | Soluble JDI TCR binding tested against HLA-A*11:01 presented KRAS^G12D peptide, mimetic peptides and random peptide pool using surface plasmon resonance**

| Protein | Peptide | Affinity $K_D$ (µM) |
|---|---|---|
| KRAS^G12D | VVVGADGVGK | 63 |
| KRAS^WT | VVVGAGGVGK | NB |
| KRAS^G12C | VVVGACGVGK | NB |
| ERAS | VVVGASGVGK | NB |
| RRAS | VVVGAGGGVGK | NB |
| MRAS | VVVGADGGVGK | NB |
| DIRAS2 | VVFGAGGVGK | NB |
| RAP1A | VVLGSGGVGK | NB |
| A11 pool1 | 10 x 10mers | NB |
| A11 pool2 | 10 x 10mers | NB |
| A11 pool3 | 8 x 9mers | NB |

NB; underlined residues indicate changes compared to KRAS^G12D peptide.
*NB* no binding.

mutation T31A, was positioned near peptide residues G4 and A5, and HLAα2 residue Q155 (Supplementary Fig. S4c, d). CDR2α mutation F54W stacked on top and formed a H-bond with HLA α2 residue Q155 and made extensive contacts with other HLAα2 residues (H151, E154 and A158) (Supplementary Fig. S4e, f). The CDR3α A100D side chain pointed away from the HLA surface and did not noticeably alter interactions with the pHLA complex. The remaining mutations contributed to either intra-loop or inter-loop interactions including in the TCRα-TCRβ variable interface. These mutations also potentially impacted pHLA binding through stabilisation of CDR loops and long-range network interactions. The buried surface area (BSA) on the KRAS^G12D peptide changed from 220 Å² (JDI TCR) to 213 Å² (JDIa41b1 TCR), and BSA on HLA changed from 770 Å² (JDI TCR) to 782 Å² (JDIa41b1 TCR), revealing marginally increased HLA surface coverage by the JDIa41b1 TCR (Supplementary Table S2).

## The JDIa41b1 TCR binds to the KRAS^WT/G12D peptides via an induced fit

To further investigate the mechanism underpinning the affinity window between JDIa41b1 TCR binding to HLA-A*11-KRAS^G12D (PDB 7OW4) versus HLA-A*11-KRAS^WT (PDB 7OW3), we solved the structures of both pHLAs without a bound TCR at 2.46 Å and 1.84 Å resolution, respectively (Table 3). Unlike the conformation adopted in the TCR bound structures, both KRAS^WT and KRAS^G12D peptides in the pHLA structures were seen in a conformation where residues 4–6 form the central bulge (Supplementary Fig. S5a, b). In the HLA-A*11-KRAS^WT structure, electron density for the central region of the peptide (residues A5 and G6) was only partially observed (Supplementary Fig. S5c), suggesting disorder likely caused by peptide mobility. For HLA-A*11-KRAS^G12D, this was even more pronounced with no observable electron density in the central region of the peptide (Supplementary Fig. S5d).

When compared to the conformation of the peptide in the JDIa41b1 TCR bound form of each complex, a large conformational shift was evident, with peptide residue 6 in both KRAS^WT and KRAS^G12D switching from an upward facing exposed position (Fig. 2e) to a downward facing position (Fig. 2f). In this TCR bound conformation, both peptides appeared more ordered in the central region, both displaying clear electron density (alignment of the KRAS^WT and KRAS^G12D peptides gave RMSD values of 0.66 Å for all atoms and 0.35 Å for only Cα atoms). However, this conformational change also resulted in differences in the interaction between the KRAS^WT and KRAS^G12D peptides and the HLA-groove, with only the KRAS^G12D peptide forming a network of interactions with the HLA F-pocket (Fig. 2d).

These observations indicate that the JDIa41b1 TCR binds via an induced fit in the peptide, enabling the TCR to engage with very similar features in both the HLA-A*11-KRAS^G12D and HLA-A*11-KRAS^WT complexes. It was also apparent that the KRAS^G12D peptide could form additional bonds with the HLA groove, possibly enabling a more stable epitope when bound to the TCR. Although both peptides adopted similar conformation in TCR-bound state, we hypothesized that a decomposition of binding energetics could provide insights into the origin of the JDIa41b1 TCR's ability to discriminate between the KRAS^WT and KRAS^G12D peptides with >4000-fold difference in affinity.

## Antigen specificity is dependent on the buried KRAS^G12D D6 residue, leading to energetic differences during TCR engagement

It has previously been reported that solvation states and thermo-dynamics can play an important role in defining TCR specificity/affinity towards antigenic peptides with minimal changes in direct contacts[25,26]. To investigate whether the thermodynamic properties could provide additional clues concerning the affinity window between the JDIa41b1-HLA-A*11-KRAS^WT, and -KRAS^G12D complexes, we performed SPR over a range of temperatures (6-36 °C), and calculated the thermodynamic parameters (Fig. 3a, Supplementary Fig. S6). These data indicated that JDIa41b1 TCR interactions with both pHLAs were enthalpically driven, but with a greater favorable value for the JDIa41b1-HLA-A*11-KRAS^G12D interaction ($\Delta H = -20.3$ kcal.mol⁻¹) compared to JDIa41b1-HLA-A*11-KRAS^WT ($\Delta H = -9.6$ kcal.mol⁻¹), indicating a larger net gain in electrostatic interactions for the former. Both TCR- pHLA complexes were entropically unfavorable, but with a greater unfavorable value for the JDIa41b1-HLA-A*11-KRAS^G12D interaction ($T\Delta S = -8.1$ kcal.mol⁻¹) compared to JDIa41b1-HLA-A*11-KRAS^WT ($T\Delta S = -1.8$ kcal.mol⁻¹). These findings are consistent with the extra bonds formed between peptide residue D6 and the HLA binding groove in the JDIa41b1-HLA-A*11-KRAS^G12D structure that might introduce extra rigidity (entropically unfavorable) and extra bonds (enthalpically favorable) during binding compared to the JDIa41b1-HLA-A*11-KRAS^WT complex. Taken together, these data suggest that, for the JDIa41b1-HLA-A*11-KRAS^G12D interaction, there was a greater energetically unfavorable shift from disorder to order (usually mediated by increases in protein rigidity during binding), but a larger net increase in new interactions, enabling the JDIa41b1 to bind with much stronger affinity to KRAS^G12D compared to KRAS^WT.

To further understand the key contributors of this energetic difference, we performed molecular dynamics (MD) simulations using the JDIa41b1-HLA-A*11-KRAS^WT, and -KRAS^G12D complex structures. Molecular Mechanics Poisson−Boltzmann surface area (MMPBSA) calculations were performed using 25 × 4 ns MD simulations per structure as many short repeat simulations have improved reliability compared to one long simulation[27,28]. The binding energies (without entropy correction) for the JDIa41b1-HLA-A*11-KRAS^G12D and -KRAS^WT complexes were determined, and the difference was calculated to be 12.6 kcal.mol⁻¹, in broad agreement with the experimentally determined difference in binding enthalpy (10.8 kcal.mole⁻¹).

The binding energy was then decomposed per residue for both structures and the difference (KRAS^G12D − WT) at each residue was determined (Supplementary Table S3, Fig. 3b, c). The largest change in the contribution to the binding energy came from residue R50α, which had a difference of 2 kcal.mol⁻¹ (Supplementary Table S3, Fig. 3d). Several residues nearby also displayed substantial differences, including K94β, H100β, N101β and Q99β in the CDR3β loop, which was directly positioned above the mutant residue (G6/D6) in the peptide. K70α also displayed a large contribution (0.88 kcal.mol⁻¹) to the binding energy despite being further from the mutation site, perhaps by compensating for changes in the electrostatics around HLA residue A158, which sat below this key lysine residue. Minimal differences were seen within the hydrogen bond networks of both K70α and R50α during the molecular dynamics simulations, suggesting that these large differences were likely due to changes in electrostatic

**Table 2 | Binding affinity of parent and various affinity enhanced TCRs to HLA-A\*11:01 presented KRAS$^{WT}$ and KRAS$^{G12D}$ peptides**

| | HLA-KRAS$^{G12D}$$K_D$ (M)$^a$ | HLA-KRAS$^{WT}$$K_D$ (M)$^a$ | Affinity window$K_D^{WT}$/$K_D^{G12D}$ | HLA-KRAS$^{G12D}$$t_{1/2}$ (h) |
|---|---|---|---|---|
| JDI (parent TCR) | 6.30 ± 0.3E-05 | NB | NA | – |
| JDIa9bwt (1st generation affinity enhanced TCR) | 2.61 ± 0.03E-07 | 2.58 ± 0.07E-05 | 98 | 0.03 |
| JDIa41b1 (2nd generation affinity enhanced TCR) | 7.43 ± 0.18E-10 | 3.00E-06 | >4000 | 1.2 |
| JDIa96b35 (3rd generation affinity enhanced TCR) | 6.37 ±1.03E-11 | 4.25 ± 0.15E-07 | >6000 | >20 |

$^a$±standard deviation, ($n$ = 2)

**Table 3 | X-ray data collection and refinements statistics**

| PDB CODE<br>Molecule | 7OW3<br>HLA-A\*11-KRAS$^{WT}$ | 7OW4<br>HLA-A\*11-KRAS$^{G12D}$ | 7PB2<br>JDI TCR-HLA-A\*11-KRAS$^{G12D}$ | 7OW5<br>JDIa41b1-HLA-A\*11-KRAS$^{WT}$ | 7OW6<br>JDIa41b1-HLA-A\*11-KRAS$^{G12D}$ |
|---|---|---|---|---|---|
| Space group | P 1 21 1 | P 21 21 21 | P 41 21 2 | P 41 2 2 | P 41 2 2 |
| Unit cell dimensions | $a$ = 71.85, $b$ = 117.03, $c$ = 110.64; α, γ = 90°, β = 99.3° | $a$ = 117.22, $b$ = 121.43, $c$ = 127.86; α, β, γ = 90° | $a$ = 208.37, $b$ = 208.37, $c$ = 124.67; α, β, γ = 90° | $a$ = 145.55, $b$ = 145.55, $c$ = 120.60; α, β, γ = 90° | $a$ = 144.16, $b$ = 144.16, $c$ = 119.59; α, β, γ = 90° |
| X-ray source | DLS I03 | DLS I04-1 | DLS I04-1 | DLS I04 | DLS I04 |
| Wavelength (Å) | 0.9762 | 0.9159 | 0.9159 | 0.9795 | 0.9795 |
| Resolution range (Å) | 79.84–2.46 (2.50–2.46)$^a$ | 88.05–1.81 (1.81–1.84) | 147.34–3.41 (3.47–3.41) | 65.09–2.58 (2.62–2.58) | 64.47–2.64 (2.69–2.64) |
| Observations | 491,760 (25,584) | 2,507,376 (120,064) | 1,081,745 (32,684) | 1,104,483 (56,843) | 1,000,018 (51,330) |
| Unique reflections | 65,854 (3282) | 166,212 (8252) | 38,015 (1869) | 41,420 (2057) | 37,600 (1851) |
| Multiplicity | 7.5 (7.8) | 15.1 (14.5) | 28.5 (17.5) | 26.7 (27.6) | 26.6 (27.7) |
| Completeness (%) | 99.9 (100) | 100 (100) | 99.98 (99.52) | 100 (100) | 100 (99.89) |
| Mean $I/\sigma(I)$ | 9.3 (2.0) | 8.7 (0.7) | 18.7 (1.4) | 14.7 (0.3) | 18.1 (0.4) |
| $R_{merge}$ | 0.189 (1.147) | 0.260 (4.611) | 0.135 (2.603) | 0.151 (5.709) | 0.115 (4.760) |
| $R_{meas}$ | 0.204 (1.231) | 0.269 (4.778) | 0.138 (2.680) | 0.153 (5.816) | 0.118 (4.848) |
| $R_{pim}$ | 0.074 (0.440) | 0.069 (1.244) | 0.026 (0.622) | 0.030 (1.102) | 0.023 (0.916) |
| $CC_{1/2}$ | 0.99 (0.562) | 0.998 (0.322) | 1.0 (0.618) | 0.999 (0.343) | 1.0 (0.363) |
| **Refinement** | | | | | |
| $R_{work}$ / $R_{free}$ (%) | 22.8 / 26.0 | 22.0 / 24.9 | 20.9 (25.1) | 20.6 / 27.4 | 21.1 / 26.8 |
| RMS (bonds) | 0.011 | 0.0091 | 0.009 | 0.0094 | 0.0093 |
| RMS (angles) | 1.686 | 1.482 | 1.660 | 1.767 | 1.685 |
| **$B$ value (Å$^2$)** | | | | | |
| Overall | 35.28 | 26.14 | 86.21 | 88.0 | 95.08 |
| HLA (chain A) | 35.37 | 30.62 | 79.48 | 88.65 | 96.05 |
| Peptide (chain C) | 51.38 | 43.57 | 79.59 | 78.11 | 78.89 |
| TCR alpha (chain D) | - | - | 93.99 | 108.06 | 114.97 |
| TCR beta (chain E) | - | - | 84.32 | 87.36 | 93.91 |
| Atom count | 12705 | 13589 | 12783 | 6632 | 6536 |
| **Ramachandran statistics** | | | | | |
| Favoured (%) | 97.67 | 99.33 | 91.29 | 93.18 | 92.20 |
| Allowed (%) | 2.33 | 0.67 | 8.71 | 6.82 | 7.80 |
| Outliers (%) | 0 | 0 | 0 | 0 | 0 |

$^a$Values in the parentheses refer to the outer resolution shell.

interactions. R28α, a residue with weak side-chain density in the JDIa41b1-HLA-A\*11-KRAS$^{G12D}$ complex, favours G12D binding, but this is fully compensated for by its HLA E58 salt-bridge partner (Supplementary Table S3). Notably, analysis of H-bonding in the MD simulations indicates that the occupancy of H-bond interactions between R28α and E58 is low, especially in the JDIa41b1-HLA-A\*11-KRAS$^{G12D}$ complex (<30%), consistent with the weak side-chain density observed.

There was a considerable difference in surface electrostatic potential of HLA-A\*11-KRAS$^{G12D}$ and -KRAS$^{WT}$ around the mutation site, and the TCR interaction zone. This shift, towards a more negative electrostatic potential in HLA-A\*11-KRAS$^{G12D}$, allowed an increased contribution for these residues in the TCR due to their largely positive electrostatic potential acting as a more complementary surface (Fig. 3e, f). Of the residues that were unique in the JDIa41b1 TCR compared to either the JDI wild-type TCR or the JDIa9bwt TCR (Table 4), CDR2α residue W53 (which directly contacted peptide residue D6), CDR2α residue Pro52 and CDR3β residue Lys94 played significant energetic roles in selectively binding to HLA-A\*11-KRAS$^{G12D}$ (Supplementary Table S3).

Overall, although the structural analysis of the TCR-pHLA complexes did not reveal an obvious mechanism for the much stronger TCR affinity for HLA-A\*11-KRAS$^{G12D}$ compared to HLA-A\*11-KRAS$^{WT}$, the thermodynamics suggested a distinct energetic mechanism driven by a large increase in favourable electrostatic interactions, despite an

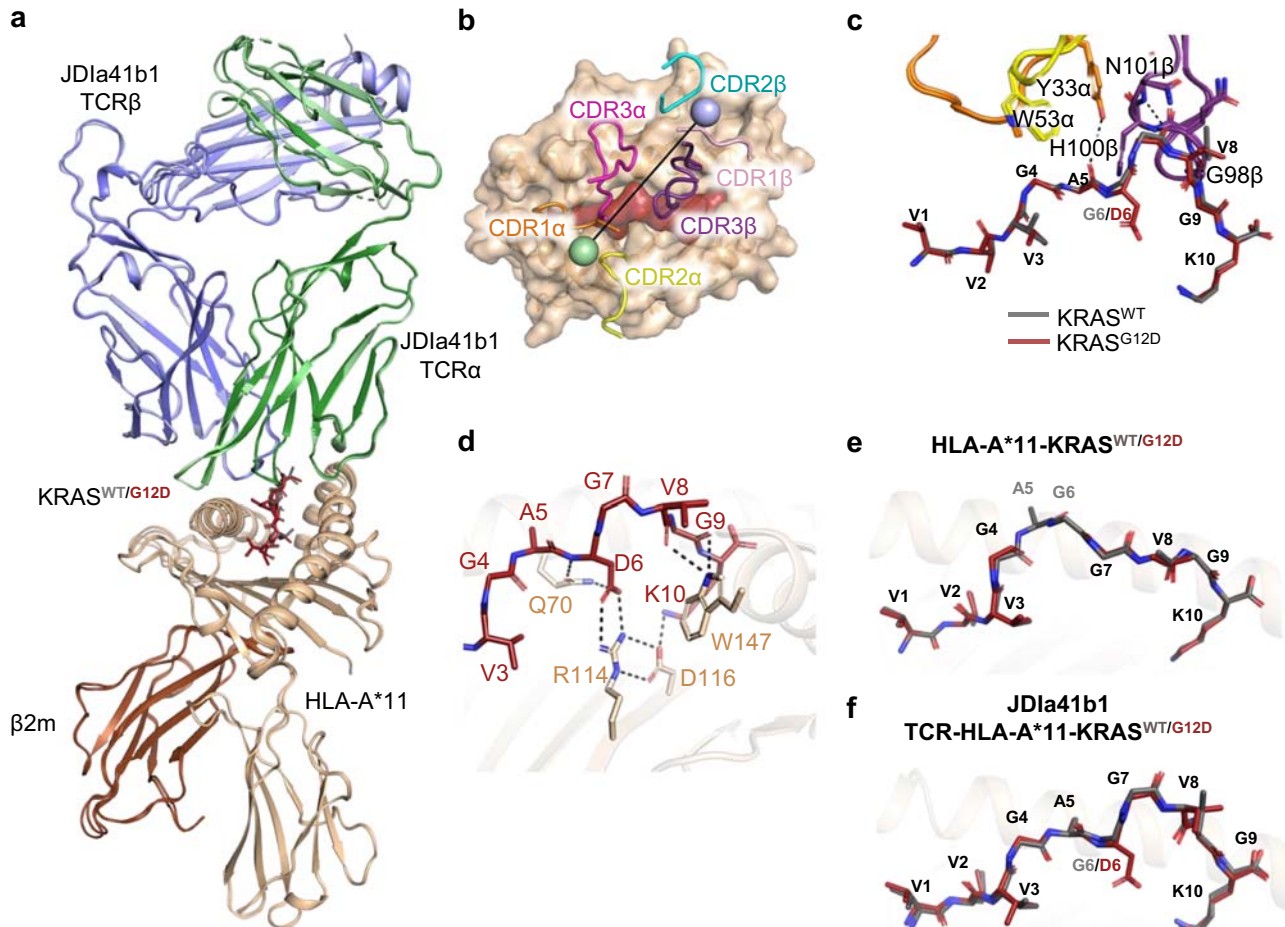

**Fig. 2 | The JDIa41b1 TCR adopts a virtually identical binding mode in complex with HLA-A*11-KRAS$^{G12D}$ and HLA-A*11-KRAS$^{WT}$. a** Superposition of JDIa41b1-HLA-A*11-KRAS$^{WT}$ (PBD 7OW5) and JDIa41b1-HLA-A*11-KRAS$^{G12D}$ (PBD 7OW6) complexes. The HLA, β2m, JDIa41b1 TCRα, and TCRβ are coloured in wheat, brown, green, and blue respectively. The darker shades of green and blue correspond to the JDIa41b1 TCR in the JDIa41b1-HLA-A*11-KRAS$^{G12D}$ complex. The KRAS$^{WT}$ and KRAS$^{G12D}$ peptides are coloured in grey and maroon respectively. **b** Top view of the JDIa41b1-HLA-A*11-KRAS$^{G12D}$ complex (PBD 7OW6). The HLA and KRAS$^{G12D}$ peptide are shown in surface representation and the CDRs are shown in cartoon tube representations. The crossing angle vector is drawn connecting the disulphides between the JDIa41b1 TCRα (green sphere) and TCRβ (blue sphere) variable domains. **c** Close up view of the JDIa41b1 TCR-peptide interactions (as overlayed in **a**). The dotted lines indicate polar contacts. **d** The HLA interaction network around the peptide residue D6 in the JDIa41b1-HLA-A*11-KRAS$^{G12D}$ complex (PBD 7OW6). **e** Superposition of the HLA-A*11-KRAS$^{WT}$ (PBD 7OW3) and HLA-A*11-KRAS$^{G12D}$ (PBD 7OW4) complexes without a bound TCR showing the peptides adopt open conformations. **f** Superposition of TCR bound JDIa41b1-HLA-A*11-KRAS$^{WT}$ (PBD 7OW5) and JDIa41b1-HLA-A*11-KRAS$^{G12D}$ (PBD 7OW6) complexes showing the peptides adopt closed conformations.

**Table 4 | CDR sequences and binding affinities of TCRs used in structural analysis**

| TCR | αCDR1 | αCDR2 | αCDR3 | βCDR2 | βCDR3 | KRAS$^{G12D}$ $K_D$ (M) | KRAS$^{WT}$ $K_D$ (M) | Affinity window $K_D^{WT}$/$K_D^{G12D}$ |
|---|---|---|---|---|---|---|---|---|
| JDI (parent TCR) | TRDTTYY | RNSFDEQNE | ALSGPSGAGSYQLT | SVGEGT | ASSYGPGQHNSPLH | 6.30E-05 | NB | NA |
| JDIa41b1 (high affinity TCR) | TRDTAYY | QPWWGEQNE | AMSVPSGDGSYQFT | SVGEGT | ASKVGPGQHNSPLH | 7.43E-10 | 3.00E-06 | >4000 |

energetically unfavourable shift requiring a larger disorder-order transition for complex formation. The molecular dynamics simulations demonstrated that these changes were directed by the KRAS$^{G12D}$ mutation, enabling energetic benefits in the TCR-pHLA interaction due to improved electrostatic complementarity between the TCR paratope and the pHLA surface.

### Affinity-enhanced JDIa96b35 TCR binds to KRAS$^{G12D}$-pHLA with high specificity

Affinity enhancement of TCRs can alter the interface with pHLA. Ensuring that the introduced mutations maintain or even enhance selectivity, relative to other peptides in the self-repertoire, is therefore of critical importance. To expand our assessment of JDI TCR specificity further than KRAS$^{WT}$, we selected a third generation affinity-enhanced JDIa96b35 TCR (Fig. 1e) (HLA-A*11-KRAS$^{G12D}$ $K_D$ 64 ± 10 pM; HLA-A*11-KRAS$^{WT}$ $K_D$ 425 ± 15 nM; affinity window >6000) and panned against an HLA-A*11 pHLA library, utilising a widely adopted single chain trimer format displayed on phage to generate a peptide specificity profile[29] (Fig. 4a). The HLA-A*11 pHLA library encodes peptide diversity at the DNA level with approximately $6.6 \times 10^9$ variants. (Fig. 4b). Following three rounds of panning, 452 unique peptides with an isolation count >1 and containing a canonical HLA-A*11 anchor (position 10 R/K) were identified and used to generate a peptide specificity profile (Fig. 4c). Consistent with the sequence of the KRAS$^{G12D}$ peptide, the 5 most

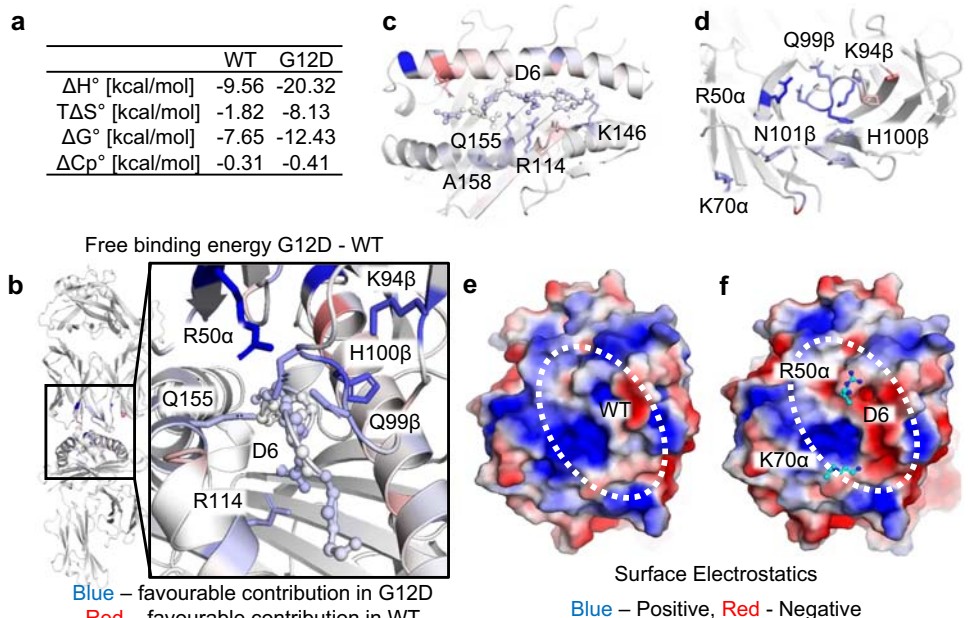

**Fig. 3 | Difference in binding energy is primarily driven by changes in electrostatic potential in pHLA mediated by the D6 mutation. a** Enthalpy, entropy, and Gibb's free energy values obtained from thermodynamic analysis. **b–d** Color mapping of the differences between JDIa41b1-HLA-A*11-KRAS$^{G12D}$ and -HLA-A*11-KRAS$^{WT}$ (G12D – WT) in terms of the contribution to the binding energy per residue, showing the residues with higher contribution to the binding energy in G12D (blue) and those with higher contribution in the WT (red). The scale is ±2 kcal.mol-1. The peptide is shown as ball and sticks throughout and any residues with significant differences ($P < 0.01$) over ±0.25 kcal.mol$^{-1}$ are shown as sticks. Key residues are labelled. **b** TCR-pHLA complex with a zoom in on the main changes in the contribution to the binding energy between the WT and G12D peptide bound complexes shown in the box. **c** pHLA top-down view. **d** TCR top-down view. **e** Surface electrostatics of HLA-A*11-KRAS$^{WT}$ with the TCR binding zone indicated by dotted white circle. **f** Surface electrostatics of HLA-A*11-KRAS$^{G12D}$ with the TCR binding zone indicated by dotted white circle and key TCR residues in terms of energetic contribution shown as cyan sticks. **e, f** Surface electrostatics based on static structures prepared for simulation. The scale used is ±2 eV.

enriched peptides from the library panning (which accounted for 96% of all the sequenced peptides) all shared the same GADG motif at residues 4 to 7 (Table 5). Next, we sought to identify peptides in the self-repertoire that are consistent with this specificity profile and may act as structural mimetics of the KRAS$^{G12D}$ peptide. 10 additional peptides were identified based on the amino acid preference across the full peptide length (Table 5). This panel of peptides was then used to assess JDIa96b35 TCR binding to peptide-HLA-A*11 complexes other than HLA-A*11-KRAS$^{G12D}$ using SPR. No measurable binding was detected to 18/20 of these pHLAs. Two peptides, derived from *RASL10A* and *DNHD1*, bound with weak affinity of 460 nM and 2.3 μM, respectively, indicating a high level of selectivity towards KRAS$^{G12D}$ (Table 5, Supplementary Fig. S7).

In order to investigate the specificity and potency of the JDI TCR in the context of a soluble T cell redirecting molecule, we generated a bispecific ImmTAC molecule consisting of the affinity-enhanced JDIa96b35 TCR fused to a humanized anti-CD3-specific scFv (referred to hereafter as IMC-KRAS$^{G12D}$, Fig. 1e). ImmTAC molecules function by binding directly to their pHLA target presented on cancer cells, thereby activating and redirecting T cells to target tumor cells[21,22,30]. IFNγ ELISPOT assays were used to measure the activation of unstimulated PBMC effectors when incubated with SUP-B15 cells loaded with decamer KRAS$^{G12D}$, KRAS$^{WT}$ or GADG motif peptides in the presence of 100 pM IMC-KRAS$^{G12D}$.

Consistent with the results of the SPR experiments, significant IFNγ release was detected in co-cultures incubated with KRAS$^{G12D}$ peptide. IFNγ release was also detected in co-cultures incubated with KRAS$^{WT}$ peptide as well as peptides derived from the proteins encoded by *RASL10A* & *DNHD1* containing the GADG motif, albeit at levels far below that observed for KRAS$^{G12D}$ peptide (equivalent to approximately 1% (KRAS$^{WT}$ and *RASL10*) and 12% (*DNHD1*) of response to KRAS$^{G12D}$) (Fig. 4d).

In IFNγ dose-response experiments with cells presenting supraphysiological levels of peptide, IMC-KRAS$^{G12D}$ produced half-maximal effective concentration (EC$_{50}$) values of 4.6 ± 2.8 pM on cells loaded with KRAS$^{G12D}$ peptide, consistent with the $K_D$ values for the corresponding TCR-pHLA interaction. We also detected CD25 + CD69 + T cells in co-cultures of PBMC effectors and SUP-B15 cells incubated with decamer KRAS$^{G12D}$ peptide and IMC-KRAS$^{G12D}$, but not in co-cultures lacking KRAS$^{G12D}$ peptide, confirming T cell activation (supplementary Fig S8). In contrast, cells incubated with peptides derived from KRAS$^{WT}$, *RASL10A* & *DNHD1* generated mean EC$_{50}$ values of 2409 pM, 1786 pM and 162.5 pM respectively. Importantly, no IFNγ response was elicited from SUP-B15 cells co-cultured with any of the other 18 GADG motif peptides tested (exemplified by TCP1 peptide in Fig. 4e). Taken together, these data indicate IMC-KRAS$^{G12D}$ has a high degree of specificity to exogenous KRAS$^{G12D}$ pHLA presented on the cell surface.

### IMC-KRAS$^{G12D}$ mediated T cell activation and redirected killing of cancer cells expressing KRAS$^{G12D}$, but not KRAS$^{WT}$

To further establish the epitope specificity of IMC-KRAS$^{G12D}$, IFNγ ELISPOT assays with professional antigen presenting cells and purified pan T cells were used to measure the activation of T cells in vitro. Immature dendritic cells (iDC) were differentiated from PBMC from two healthy HLA-A*11+ donors, electroporated with mRNA containing genes encoding for either KRAS$^{WT}$ or KRAS$^{G12D}$, and co-cultured with autologous T cells and IMC-KRAS$^{G12D}$ (Fig. 5a). Addition of IMC-KRAS$^{G12D}$ to cultures resulted in a concentration dependent IFNγ response to cells electroporated with mRNA encoding KRAS$^{G12D}$. In contrast, no IFNγ release was observed from co-cultures of iDC transfected with KRAS$^{WT}$. Because multiple peptides encompassing the G12 region of KRAS have been reported[20], untransfected iDC were pulsed with KRAS$^{WT}$ and KRAS$^{G12D}$ nonamer and decamer peptides

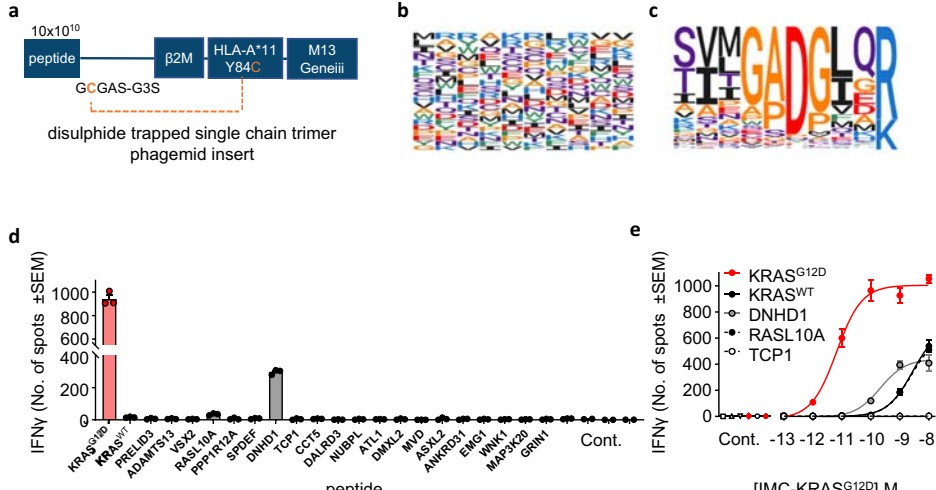

**Fig. 4 | Peptide library screening shows that high affinity TCR JDIa96b35 has strong preference for D6 residue of HLA-A*11-KRAS[G12D]. a** Schematic representation of the disuphide trapped single chain pHLA trimer construct. **b** The amino acid composition of the fully randomized peptide library. **c** Peptide specificity profile generated from the 452 peptides with $n > 1$ following three cycles of panning with the TCR JDIa96b35. **d** IFNγ ELISPOT output after SUP-B15 cells were pulsed with 10 μM indicated peptide, treated with 100 pM IMC-KRAS[G12D] and co-cultured with PBMC for 24 h. Targets alone, PBMC alone, no IMC-KRAS[G12D], and no peptide negative controls (Cont.) were performed. Mean data of three biological replicates ±SEM from one representative experiment from two independent experiments is shown. **e** IFNγ ELISPOT output after SUP-B15 cells were pulsed with 10 μM KRAS[G12D], KRAS[WT], *DNHD1*, *RASL10A*, or TCP1-derived peptide, treated with IMC-KRAS[G12D] and co-cultured with PBMC for 24 h. Targets alone, PBMC (alone, with IMC-KRAS[G12D] and no peptide or with IMC-KRAS[G12D] and peptide), no IMC-KRAS[G12D] and no peptide negative controls (Cont.) were performed. Mean data of three biological replicates ±SEM from one representative experiment from two independent experiments is shown. Source data are provided as a Source Data file.

(KRAS VVGAGGVGK, VVGADGVGK, VVVGAGGVGK, VVVGADGVGK) and co-cultured with naïve autologous T cells in the presence of IMC-KRAS[G12D]. As expected, IMC-KRAS[G12D] mediated T cell activation in co-cultures containing iDC that were pulsed with the KRAS[G12D] decamer, but not with either of the KRAS[WT] decamer, KRAS[G12D] nonamer or the KRAS[WT] nonamer peptides (Fig. 5b).

To further determine the specificity and biological activity of IMC-KRAS[G12D], HLA-A*11+ tumor cell lines naturally presenting KRAS[G12D] mutations were investigated. KRAS[G12D] mutations are present in tumors from a diverse range of lineages but are enriched in tumors as well as in cell lines derived from patients with pancreatic and colon adenocarcinoma[31] (cancer.sanger.ac.uk). However, only a limited number of cell lines express both KRAS[G12D] and HLA-A*11, and several of these HLA-A*11+ pancreatic KRAS[G12D] + cell lines express only low levels of HLA or antigen processing machinery[32]. Consistent with observations from studies of T cells modified to express murine TCR specific for KRAS[G12D] [20], we only observed T cell activation in the presence of IMC-KRAS[G12D] when these cells were modified to overexpress HLA-A*11 (see PANC-1 HLA-A*11 cells in Fig. 5e). Therefore, CRISPR/Cas9 editing was used to introduce DNA encoding either the KRAS[G12D] mutation or HLA-A*11 into the natural locus under the control of the native promoter of a pancreatic adenocarcinoma cell line PSN-1. We confirmed the presence of DNA sequence encoding the KRAS[G12D] mutation by sequencing the *KRAS* gene and used western blot analysis and specific antibodies to confirm the expression of KRAS[G12D] (Fig. 5c), while expression of HLA-A*11 was confirmed by HLA-typing and flow cytometry with an HLA-A*11 specific antibody. The HLA- and KRAS-edited isogenic cell lines were then used as a tool to dissect the specificity of IMC-KRAS[G12D] in a comparable and consistent cellular background. IMC-KRAS[G12D] mediated IFNγ release was observed in ELISPOT assays using PBMC co-cultured with PSN-1 cells modified to express both KRAS[G12D] and HLA-A*11 simultaneously (EC50 67.6 ± 15 pM & 136 ± 52 pM for clone 1 & 2 respectively), but not PSN-1 cells expressing only KRAS[G12D] and parental HLA-A*24 (clone 4) or KRAS[WT] and HLA-A*11 (clone 3) (Fig. 5d). Taken together, these data indicate IMC-KRAS[G12D] is

specific for KRAS[G12D] naturally presented in the context of HLA-A*11 on the cancer cell surface.

The ability of IMC-KRAS[G12D] to specifically mediate T cell activation and redirect T cell killing of cancer cells expressing natural KRAS[G12D] was determined using IFNγ ELISPOT assays and a quantitative live cell imaging assay. HLA-A*11+ cancer cell lines CL40 (KRAS[G12D]), SK-Mel-28 (KRAS[WT]) and NCI-H2030 (KRAS[G12C]) were utilized, as well as a panel of normal cells including pulmonary fibroblasts, cardiac myocytes, cardiac smooth muscle cells, aortic endothelial and colon epithelial cells isolated from healthy donors. Incubation of CL40 colon cancer cells with IMC-KRAS[G12D] together with HLA-A*11+ PBMC resulted in IMC-KRAS[G12D] concentration dependent IFNγ, IL-2 and Granzyme B release as well as redirected T cell killing in a time- and dose-dependent manner with a mean EC50 value of 28.5 ± 25 pM (Fig. 5e, f and supplementary Fig S9). IMC-KRAS[G12D] treatment resulted in little to no IFNγ release in co-cultures of PBMC & HLA-A*11+ cell lines SK-Mel-28 (KRAS[WT]), SUP-B15 (KRAS[WT]) or NCI-H2030 (KRAS[G12C]). Significantly, these cell lines express both *KRAS* and *DNHD1*, sources of potential mimetic peptides, at physiological levels (supplementary Table S5). Incubation of KRAS[G12D] negative cell lines with KRAS[G12D] peptide elicited a substantial response above background in the presence of IMC-KRAS[G12D], confirming the integrity of these cells and their ability to present peptides (Fig. 5e). Consistent with the IFNγ ELISPOT assays, no IMC-KRAS[G12D] mediated T cell killing was observed in co-cultures of SK-Mel-28 or NCI-H2030 with HLA-A*11+ PBMC (Supplementary Fig. S9). Most importantly, IMC-KRAS[G12D] did not elicit IFNγ release when incubated with pulmonary, cardiac or colon-derived normal cells isolated from healthy volunteers (Fig. 5e) or result in T cell mediated targeting of normal colonic epithelial cells (KRAS[WT]), in contrast to cell lines expressing KRAS[G12D] (Fig. 5e, f). Taken together, these data suggest that the affinity-enhanced, soluble bispecific ImmTAC molecule IMC-KRAS[G12D], is a potent and specific T cell engager capable of mediating T cell activation and redirected T cell targeting of cancer cells harboring KRAS[G12D] but not cancer or normal cells expressing KRAS[WT].

**Table 5 | Mimetic peptides selected from the human genome based on the binding motif determined using scHLA phage library screening**

| | | Peptide amino acid residue number: | | | | | | | | | 1-10 | 4-7 | | |
| | | 1 | 2 | 3 | 4 | 5 | 6 | 7 | 8 | 9 | 10 | | | | |
| UniProtID | Gene | | | | | | | | | | | Peptide Score | Motif Score | NetMHC pan4.1 | $K_D$ (M) |
|---|---|---|---|---|---|---|---|---|---|---|---|---|---|---|---|
| P01116 | KRAS[G12D] | V | V | V | G | A | D | G | V | G | K | 0.67 | 0.92 | 194.20 | 6.40E-11 |
| P01116 | KRAS[WT] | V | V | V | G | A | G | G | V | G | K | 0.58 | 0.69 | 153.55 | 4.30E-07 |
| Q96N28 | PRELID3A | S | V | L | G | V | D | V | L | Q | R | **0.68** | 0.53 | 132.59 | - |
| Q76LX8 | ADAMTS13 | S | V | S | C | G | D | G | I | Q | R | **0.66** | 0.61 | 567.49 | - |
| P58304 | VSX2 | T | V | S | G | P | D | S | L | A | R | **0.65** | 0.65 | 333.05 | - |
| Q92737 | RASL10A | A | V | L | G | A | P | G | V | G | K | **0.63** | 0.73 | 23.04 | 4.60E-07 |
| O14974 | PPP1R12A | T | V | T | S | A | A | G | L | Q | K | **0.63** | 0.51 | 44.08 | - |
| O95238 | SPDEF | A | A | A | G | A | V | G | L | E | R | **0.63** | 0.69 | 625.61 | - |
| Q96M86 | DNHD1 | T | V | L | G | P | N | G | V | G | K | **0.63** | 0.65 | 31.32 | 2.30E-06 |
| P17987 | TCP1 | S | S | L | G | P | V | G | L | D | K | **0.62** | 0.64 | 18.46 | - |
| P48643 | CCT5 | T | S | L | G | P | N | G | L | D | K | **0.62** | 0.65 | 84.28 | - |
| Q5D0E6 | DALRD3 | T | V | L | V | A | D | H | L | A | R | **0.62** | 0.50 | 603.65 | - |
| Q8TB37 | NUBPL | H | I | F | G | A | D | G | A | R | K | 0.49 | **0.92** | 284.91 | - |
| Q8WXF7 | ATL1 | F | S | Y | G | A | D | G | G | A | K | 0.47 | **0.92** | 260.49 | - |
| Q8TDJ6 | DMXL2 | F | S | C | G | A | D | G | T | L | K | 0.50 | **0.92** | 343.48 | - |
| P53602 | MVD | H | L | L | G | P | D | G | L | P | K | 0.56 | **0.87** | 119.04 | - |
| Q76L83 | ASXL2 | R | Q | V | G | P | D | G | L | M | K | 0.57 | **0.87** | 123.42 | - |
| Q8N7Z5 | ANKRD31 | L | L | N | G | A | D | P | L | F | R | 0.54 | **0.84** | 883.91 | - |
| Q92979 | EMG1 | S | V | R | A | A | D | G | P | Q | K | 0.59 | **0.81** | 140.41 | - |
| Q9NYL2 | MAP3K20 | V | V | I | A | A | D | G | V | L | K | 0.60 | **0.81** | 32.48 | - |
| Q9H4A3 | WNK1 | V | S | V | A | A | D | G | A | Q | K | 0.57 | **0.81** | 172.60 | - |

'–' Indicates no binding.

## Discussion

Recent evidence has demonstrated that it is possible to directly target cancer neoantigens via the pHLA pathway[3,4,19]. In this study, we report the identification of a TCR that specifically recognizes a KRAS[G12D] neoantigen peptide presented in the context of HLA-A*11. Neoantigens represent the opportunity to target cancer cells with pHLA-specific therapeutics with a low risk of off-tumor toxicity providing selectivity for the mutated epitope is retained.

Crystal structures have provided some mechanistic understanding of neoantigen specificity. The aspartate residue of the KRAS[G12D] peptide presented by HLA-C*08:02 (GADGVGKSAL) was reported to form a salt bridge with HLA residue R156 to act as an anchor. Glycine in the wild-type peptide at position 3 is not an optimal anchor residue and hence is not effectively presented by HLA-C*08:02. In this case, poor wild-type peptide presentation was likely the mechanism underlying TCR specificity towards KRAS[G12D] mutant peptide[33,34]. A similar mechanism, whereby somatic mutations introduce preferred primary HLA anchor residues, has been shown to generate public, shared neoantigens, for example in PIK3CA mutations encoding PI3Kα[H1047L] and histone variant H3.3[K27M] [35,36]. In contrast, both KRAS[WT] and KRAS[G12D] peptides used in this study had comparable predicted binding affinities to HLA-A*11, suggesting that selectivity based on peptide presentation was unlikely to occur. Despite this, biochemical analysis of the JDI TCR demonstrated a preference for KRAS[G12D] and it was possible to enhance the binding affinity of this TCR

to HLA-A*11-KRAS[G12D] by over a million-fold, while conserving the ability to distinguish between KRAS[WT] and KRAS[G12D].

Structural analysis of an affinity-enhanced version of the JDI TCR in complex with HLA-A*11-KRAS[WT] and -KRAS[G12D] demonstrated a virtually identical interaction network. However, analysis of the pHLA crystal structures revealed that both peptides underwent an induced fit conformational change upon TCR binding, with peptide residue D6 in the HLA-A*11-KRAS[G12D] complex forming a network of stabilizing interactions with the HLA-A*11 F-pocket upon TCR binding. This change in peptide conformation was reflected by thermodynamic analysis, demonstrating that TCR binding to HLA-A*11-KRAS[G12D] was driven by a greater net formation of new electrostatic interactions, at the cost of a greater disorder-order transition compared to HLA-A*11-KRAS[WT]. Further analysis using molecular dynamics simulations suggested that the buried D6 residue mediated a change towards negative electrostatics on the HLA surface, increasing the electrostatic interactions with the TCR. This allowed energetically favourable electrostatic interactions, in part mediated by TCR residues R50α and K70α, that formed an indirect network of interactions with the pHLA surface[37]. The molecular dynamics simulations also demonstrated greater energetically favorable interactions via direct interactions with peptide residue D6 with both the HLA-A*11 F-pocket and residues in the TCR CDR3β[38–40]. The structural, dynamic and electrostatically-driven specificity signature of the high affinity JD1 TCR resembles the specificity signature of anti-EGFR antibodies that differentially recognize

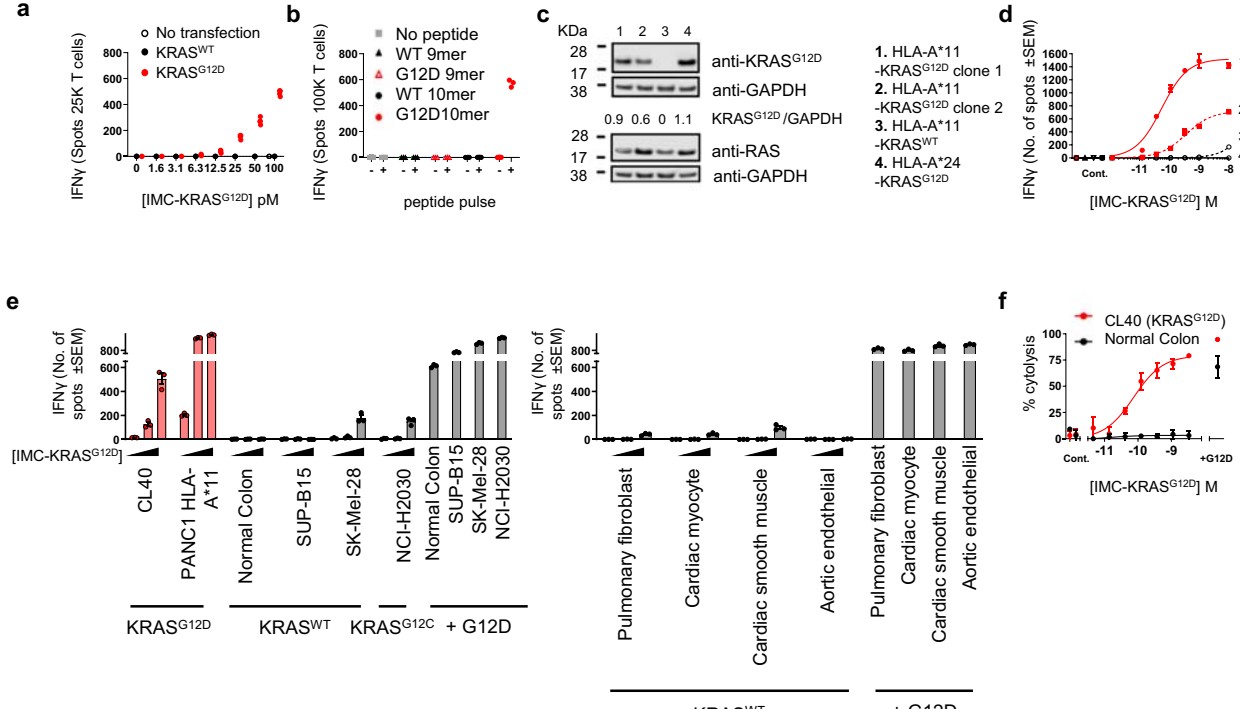

**Fig. 5 | IMC-KRAS^G12D mediated T cell activation and redirected killing of cancer cells expressing KRAS^G12D, but not KRAS^WT.** **a** Immature dendritic cells differentiated from monocytes isolated from healthy donors were transfected with mRNA encoding KRAS^WT or KRAS^G12D or **b** pulsed with indicated exogenous peptide, were treated with IMC-KRAS^G12D and co-cultured with autologous T cells for 24 h. T cell activation was measured by IFNγ ELISPOT assay. Three biological replicates from one representative experiment from two independent experiments are shown. **c** Immunoblot and densitometry analysis of isogenic PSN-1 cell lysates. Blot is representative of two independent experiments. Uncropped blots can be viewed in Supplementary Fig. S10. **d** IFNγ ELISPOT output of cell lines modified to express HLA-A*11 and KRAS^G12D (clones 1 & 2), HLA-A*11 and KRAS^WT (clone 3) or only KRAS^G12D (clone 4) treated with IMC-KRAS^G12D and co-cultured with PBMC for 24 h. Targets alone, or no IMC-KRAS^G12D negative controls (Cont.) were performed. Mean data of three biological replicates ±SEM from one representative experiment from

three independent experiments is shown. **e** IFNγ ELISPOT output of an extended panel of HLA-A*11-KRAS^G12D + and HLA-A*11-KRAS^G12D- cancer cell lines and healthy cells, treated with IMC-KRAS^G12D and co-cultured with PBMC for 24 h. HLA-A*11 + / KRAS^G12D- cell lines were pulsed with 10 μM KRAS^G12D ('+G12D') peptide as a positive control. Mean data of three biological replicates ±SEM from one representative experiment from two independent experiments is shown **f**. Redirected T cell killing assay of HLA-A*11 + /KRAS^G12D CL40 cancer or HLA-A*11 + /KRAS^WT normal human colonic epithelial cells expressing nuclear-restricted mKATE2, treated with IMC-KRAS^G12D and co-cultured with PBMC for 72 h. Cells were pulsed with 10 μM KRAS^G12D ('+G12D') peptide as a positive control. Targets alone, PBMC alone or no IMC-KRAS^G12D negative controls (Cont.) were performed. Mean data of three biological replicates ±SD from one representative experiment from two independent experiments is shown. Source data are provided as a Source Data file.

the EGFR^S468R mutation that was clinically associated with acquired cetuximab resistance in colorectal cancers[41]. These findings add further to our understanding of how TCRs can use indirect energetically driven mechanisms to sense differences not only in peptide sequence, but also in peptide dynamics, that can drive T cell antigen specificity[42].

The affinity enhanced TCR bispecific protein tebentafusp (CD3 × gp100), has been shown to redirect polyclonal T cells to specifically target a peptide derived from the intracellular protein gp100[22] and provide a survival benefit in metastatic uveal melanoma[43] without treatment-related mortality. In contrast, in the context of an adoptive cell therapy, an engineered TCR directed to MAGE-A3 pHLA was associated with off-target cardiac toxicity and patient death. Subsequent analyses revealed this MAGE-A3 TCR cross reacts with a peptide derived from the muscle protein Titin[44]. These examples demonstrate both the promise and the potential danger of engineered TCR therapies and highlight the importance of extensive molecular and functional understanding of TCR specificity prior to initiating clinical studies. Using a peptide library approach, we demonstrated that an affinity-enhanced version of the JDI TCR a96b35, retains a strong preference for D6 residue of HLA-A*11-KRAS^G12D and maintains a wide specificity window towards potential mimetics suggesting reduced risk of off-target cross reactivity.

Using functional cell-based assays, we demonstrated that an ImmTAC molecule containing the affinity-enhanced JDI TCR, IMC-

KRAS^G12D, exhibited potent and selective binding to HLA-A*11-KRAS^G12D in the context of both peptide pulsing, and natural presentation on both dendritic and cancer cells. Further, using a novel isogenic pancreatic cell line model, we confirmed that KRAS^G12D pHLA is both necessary and sufficient for IMC-KRAS^G12D-mediated T cell activation, as only cells expressing both KRAS^G12D and HLA-A*11 elicited IFNγ responses. Finally, we established IMC-KRAS^G12D-redirected targeting of colonic cancer cells naturally presenting KRAS^G12D, whilst sparing both cancer cells with comparable levels of KRAS^WT and a panel of normal cells including a range of cardiac cell types and colon epithelial cells. We believe this is the first example of a soluble T cell redirecting TCR reagent capable of targeting the most frequent cancer associated oncogenic driver mutation, KRAS^G12D, with the potential to treat patients with high unmet need.

These data, combined with other investigations[33,45,46], demonstrate that TCRs can detect single amino acid mutations in common neoantigens derived from oncogenic mutations. Here, we elaborate on the molecular mechanisms guiding TCR specificity, demonstrating that indirect energetically driven factors could result in >4000-fold affinity window between a wild-type and mutated KRAS^G12D neoantigen. This window translated into a mechanism by which a T cell engaging ImmTAC molecule bearing the affinity-enhanced TCR could achieve fine specificity for antigen positive cancer cells, whilst remaining tolerant to antigen negative cells. These findings demonstrate the powerful utility

of TCR therapeutics to target neoantigens and have important implications for understanding the molecular determinants of therapeutic TCR selectivity for neoantigens.

## Methods

### Primary Cell and cell line culture, antibodies and reagents

Cell lines and normal cells were purchased from the suppliers listed in supplementary table S6 and cultured in media recommended by the supplier. *KRAS* (exon 2 region) was sequenced from PCR products amplified from gDNA isolated using QIAamp DNA Mini Kit (QIAGEN, Hilden, Germany) at Eurofins Genomics (Ebersberg, Germany).

PBMCs were either purchased cryo-frozen from commercial suppliers (Cellular Technology Limited (CTL), OH & Tissue Solutions, Glasgow, UK) or prepared from healthy donor leukopaks (ALLCELLS, CA) by Ficoll density gradient centrifugation. CD14 + monocyte cells isolated by negative selection from PBMCs using Classical Monocyte Isolation Kit (Miltenyi, North Rhine-Westphalia, Germany) were differentiated for 4–6 days in Gibco™ AIM V Medium (Thermo Fisher Scientific, MA) with Penicillin-Streptomycin (Gibco™), 5% AB serum (Valley Biomedical, VA), 1000 U/ml GM-CSF and 500 U/ml IL-4 (Biotechne, MN) to generate iDC. Where relevant, T cells were isolated from PBMC by immunopurification using a Pan T Cell Isolation Kit (Miltenyi) according to the manufacturer's instructions. Rabbit anti-RAS, –RAS$^{G12D}$ and HRP-conjugated secondary antibodies were purchased from Cell Signaling Technology (Danvers, MA). Anti-GAPDH was purchased from Merck Millipore (MA). Purified synthetic peptides were purchased from Peptide Protein Research Ltd, UK

### T cell receptor isolation, affinity maturation and protein production

A TCR specific to HLA-A11-KRAS$^{G12D}$ was isolated from a healthy donor using established methods[47]: T cells were isolated from a HLA-A*02:01/A*11+ donor and stimulated for 7 days with autologous dendritic cells that had been pulsed with 1 μM KRAS$^{G12D}$ peptide (VVVGADGVGK) to displace naturally presented peptides. For this, the dendritic cells were incubated with exogenous peptide for two hours, then washed two times with culture media. T cells were subsequently stimulated twice more over an additional 14-day period with autologous activated B cells that had been pulsed with 0.1 μM KRAS$^{G12D}$ peptide. T cell cultures were screened by IFNγ ELISPOT assay to identify KRAS$^{G12D}$ peptide specific T cells. T cell clones were sorted on the basis of the expression of the activation markers CD25 and CD137 after incubation with the KRAS$^{G12D}$ peptide by using a FACSAriaII (BD Biosciences). TCR gene sequences were identified from a specific T cell clone by rapid amplification of cDNA ends (RACE).

TCR affinity maturation was performed using phage display methodology[23,24]. Briefly, NNK primer driven mutations were introduced into CDR segments of TCRα and TCRβ genes and phage libraries were constructed by overlap extension PCR. Wild-type HLA-A*11-KRAS was used for negative selection of phage libraries prior to selection on HLA-A*11-KRAS$^{G12D}$ to facilitate identification of mutant peptide selective clones. Three rounds of selection were performed with decreasing antigen concentration to enrich clones with enhanced affinity. Mutants were screened using competitive inhibition ELISA to select specific affinity enhanced clones. To produce soluble disulfide-linked mTCRs and IMC-KRAS$^{G12D}$ molecules, TCRα and TCRβ chain inclusion bodies were denatured using 50 mM Tris buffer pH 8.1 containing 100 mM Sodium Chloride, 6 M Guanidine and 20 mM Dithioreitol (DTT) and refolded by diluting to final protein concentration of 60 mg/L into a buffer containing 100 mM Tris pH 8.1, 4 M Urea, 400 mM L-Arginine, 1.9 mM Cystamine and 6.5 mM Cysteamine redox couple. Refolding mixture was dialysed against water followed by 20 mM Tris buffer pH 8.1. Correctly folded protein was purified by anion exchange chromatography, cation exchange chromatography and size exclusion as previously described[48].

### Binding affinity and thermodynamic parameter measurement

pHLA complexes were refolded as previously reported[49]. The HLA heavy chain AviTag™ was biotinylated using BirA500 biotin-protein ligase kit (Avidity LLC, USA). Biotinylated pHLAs were immobilized on a Streptavidin-coated CM5 sensor chip using a Biacore T200/Biacore 8 K (Cytiva life sciences, USA). 400-500 RU of biotinylated pHLA was immobilized for multi cycle steady state analysis of weak affinity TCRs using Biacore T200 and 100-200 RU of pHLA for single cycle kinetic analysis of strong affinity TCRs using Biacore 8 K. 5000 RU of common peptide (CP) pool pHLA was used for TCR cross reactivity testing. JDI TCR affinity was measured by 10-point steady state analysis using TCR concentration ranging from 50 μM to 0.1 μM. Alanine scan of JDI TCR was performed using a 10 point titration using TCR concentration ranging from 100 μM to 0.2 μM. Multi cycle steady state equilibrium analysis data was processed using one site total least squares fit in GraphPad Prism 9.0. Single cycle kinetic analysis of high affinity TCRs was performed using 5-point titration with top concentration at least 50x higher than $K_D$. Single cycle kinetic data was processed using Biacore Insight Evaluation software. A Minimum of 2 measurements were performed to obtain $K_D$ values reported for JDI TCR, JDIa9bwt, JDIa41b1 and IMC-KRAS$^{G12D}$. Binding affinities were measured ($n = 2$) at 6 °C intervals from 6 °C-36 °C using a Biacore T200 for thermodynamic parameter determination[25,27]. Data was analyzed using the Thermodynamics wizard in Biacore T200 Evaluation Software 3.1. Nonlinear Van't Hoff plots were used to determine enthalpy (ΔH), entropy (TΔS) and Gibbs free energy (ΔG).

### Crystallisation and protein structure determination

The TCR-pHLA complexes were prepared by mixing purified TCR and pHLA at a molar ratio of 1:1.15. The TCR-pHLA and pHLA samples were concentrated to ~10 mg/ml and crystallisation trials were set up by dispensing 150 nL of protein solution plus 150 nL of reservoir solution in sitting-drop vapor diffusion format in two-well MRC crystallization plates using a Gryphon robot (Art Robbins Instruments, LLC). Plates were maintained at 20 °C in a Rock Imager 1000 (Formulatrix) storage system. Diffraction quality crystals for HLA-A*11-KRAS$^{WT}$ and HLA-A*11-KRAS$^{G12D}$ complexes were grown in the following conditions: 0.2 M Ammonium sulfate, 0.1 M Sodium cacodylate pH 6.0, 25% PEG 4000; and 0.2 M Lithium sulfate, 0.1 M Bis-Tris pH 5.5, 25% PEG 3350, respectively. Diffraction quality crystals for JDI TCR-HLA-A*11-KRAS$^{G12D}$, JDIa41b1-HLA-A*11-KRAS$^{WT}$ and JDIa41b1-HLA-A*11-KRAS$^{G12D}$ complexes were grown in the following conditions: 0.2 M Ammonium sulfate, 0.1 M Tris pH 8.5, 20% PEG 8000; 0.2 M Sodium nitrate, 20% PEG 3350; and 0.2 M Ammonium sulfate, 20% PEG 3350, respectively. X-ray diffraction data for the four samples were collected at the Diamond Light Source (Oxfordshire, UK) beamlines (HLA-A*11-KRAS$^{WT}$ – DLS I03; HLA-A*11-KRAS$^{G12D}$ – DLS I04-1; JDIa41b1-HLA-A*11-KRAS$^{WT}$ and JDIa41b1-HLA-A*11-KRAS$^{G12D}$ – DLS I04). Diffraction images for the HLA-A11-KRAS$^{WT}$ and HLA-A*11-KRAS$^{G12D}$ complexes were indexed, integrated, scaled, and merged using the autoproc[50] pipeline implementing XDS/XSCALE[51] and POINTLESS/AIMLESS[52]. Diffraction images for the JDI TCR-HLA-A*11-KRAS$^{G12D}$, JDIa41b1-HLA-A*11-KRAS$^{WT}$ and JDIa41b1-HLA-A*11-KRAS$^{G12D}$ complexes were indexed, integrated, scaled, and merged using xia2[53] automated pipeline implementing DIALS[54]. Structures were solved by molecular replacement using using PDB 4UQ2 for HLA and β2m, and a hybrid TCR generated by combining PDB 4JRX (TCR α chain) and PDB 4X6B (TCR β chain) as search models in Phaser[55]. Two separate PDBs were chosen for the TCR α and β chains as these individually have very high sequence identity (>90%) for the JDI TCR chains. The TCR β chain from PDB 4X6B was superposed onto and in the place of CA5 TCR beta chain (PDB 4JRX) to create the hybrid TCR search model. Models were built using iterative cycles of manual model building in COOT[56] and refinement using Refmac[57] in the CCP4 suite[58]. The stereochemical properties and validation of the models were assessed using the PDB-REDO[59] and Molprobity[60].

The data processing and refinement statistics are listed in Table 3. Structural figures were prepared using PyMOL (Schrödinger, LLC).

## Molecular dynamic simulations-MMPBSA Calculations

The following is a short overview of the computational methods used herein; a detailed description is provided in the Supplementary Materials. The X-ray structures were used as starting points for MD simulations in aqueous solvent with a 100 mM sodium chloride concentration. Prior to production simulations, minimization, heating, and equilibration was performed as described in Supplementary Materials. The first 1 ns of the production MD was discarded to allow for further equilibration, meaning $25 \times 3$ ns of simulation and a total of 7500 snapshots were used in each MMPBSA calculation. The MMPBSA.py.MPI script was used on Amber16[61,62]. A salt concentration of 100 mM was used (with the default dielectric). All ions were stripped from the trajectories, prior to MMPBSA calculation, and the closest 30 explicit waters to the binding interface were retained. Differences in the contribution to the binding energy were calculated as the average WT value subtracted from the average G12D value at each residue. Multiple T-tests were performed on GraphPad Prism 9 (GraphPad Software, CA) to determine the significance of the differences at the $P < 0.01$ level of significance.

## Generation of HLA-A*11 scHLA libraries

Peptide HLA libraries were generated in a single chain format with peptide-β2m- HLA-A*11 displayed on the surface of phage as disulfide trapped single chain trimers (dsSCT). Briefly a randomized 10-mer peptide library consisting of $1 \times 10^{10}$ peptide diversity was synthesized (Twist Biosciences, USA) and cloned into a phagemid scHLA construct using a pelB leader sequence and C-terminal coat protein pIII[37]. This phagemid library was introduced by electroporation into E. coli TG1 cells with KM13 helper phage to enable monovalent display with an estimated library size of $6.6 \times 10^9$ colonies[63]. Diversity was confirmed post electroporation by next generation sequencing of the initial library. This confirmed that all 20 amino acids were represented at every position with a flat distribution of 5% (max 5.8%, min 4.5%).

## Panning of scHLA libraries with JDIa41b1 TCR

Streptavidin-coated paramagnetic beads M280 (Thermo Fisher Scientific, USA) were saturated with biotinylated TCR and phage selections performed as described previously[37]. Colonies were scraped and phagemid purified by miniprep (Qiagen). Sequencing libraries were prepared by amplification of the peptide region. Purified PCR products (Ampure XP beads, Beckman coulter) were prepared using NebNext Ultra II DNA library prep kit (NEB, E7645S) and $2 \times 150$ bp paired end read amplicon sequencing performed with the Illumina MiSeq. From >0.9 million peptides sequences 96.5% had an R/K as the preferred C-terminal anchor for HLA-A*11 allele. Peptides identified with other amino acids at position 10 had low counts and no sequence convergence. This noise was removed, and peptide specificity profiles generated from 1182 unique peptides, where each peptide is presented once.

## Selection of potential crossreactive mimetic peptides

The peptide specificity profile was used to identify potential mimetics by comparing the amino acid preference at each position to the HLA allele specific human peptidome. The human proteome at UniProtKB (Proteome ID UP000005640) comprising of 42132 canonical and isoform proteins[64] was processed into 10 amino acid overlapping peptides and a theoretical allele specific repertoire was estimated using netMHCpan4.1[65]. All peptides with a predicted binding affinity to the HLA-A11*01 of weaker than $IC_{50} >1\,\mu M$ were excluded from further analysis. Each peptide in the theoretical HLA-A11*01 self-peptidome was then scored for consistency to the experimentally determined

peptide specificity profile. Scores were calculated as the normalised product of the amino acid percent frequencies at each position. The values were normalised according to the maximum possible score ($100 \times 100 \times 100...$) for a given length. Peptides with higher scores are therefore more consistent with the motif. (Table 5). Of the 10 potential mimetics with highest scores, reassuringly only 4/10 peptides lack the Asp6 residue which we have demonstrated is important for the orientation of the peptide. To ensure that potential mimetics were not overlooked we restricted the specificity profiles to the TCR binding interface, scoring only against the central peptide residues, positions 4,5,6,7. A further 10 peptides were identified and in total 20 peptides were synthesised (Peptide Protein Research Ltd, UK.) for further cross-reactivity assessment.

## IFNγ ELISPOT assay

IFNγ ELISPOT assays were performed using BD™ ELISPOT reagents in accordance with manufacturer's instructions[66]. For assays using exogenous peptide loaded cells, target cells were incubated with $10\,\mu M$ peptide (unless otherwise indicated) and co-cultured with PBMC from non-HLA*A11:01/03:01 donors in the presence of IMC-KRAS$^{G12D}$ in RPMI-1640 medium containing 25 mM HEPES and supplemented with Penicillin-Streptomycin (Gibco™) and 10% fetal bovine serum overnight at 37 °C. For assays using iDC, capped mRNA encoding KRAS$^{WT}$ or KRAS$^{G12D}$ (TriLink, CA) was electroporated into iDCs using Amaxa Human B Cell Nucleofector Kit (Lonza Switzerland) with a single electrical pulse on Amaxa Nucleofector II device (Lonza) using program X-001. The dendritic cells were allowed to recover in complete AIM-V medium at 37 °C overnight and were then harvested and co-cultured with T cells. Alternatively, iDCs were pulsed with $5\,\mu g/ml$ peptides (Bio-Synthesis, TX) for 1 h at 37 °C, and then co-cultured with T cells. Freshly isolated T cells were then co-cultured with iDCs at a 5:1 ratio and treated with IMC-KRAS$^{G12D}$ in CTL test medium (CTL) overnight at 37 °C on the 96-well Strip Precoated Human IFNγ Single-Color Enzymatic ELISPOT plate (CTL). IFNγ release was quantified using the the CTL ImmunoSpot Analyzer and ImmunoSpot® software. (Immunospot Series 5 Analyzer, CTL).

## Redirected T cell killing assays

Kinetic redirected T cell killing assays were performed using live cell imaging[67]. Briefly, target cells were transduced with commercially available lentivirus encoding mKATE2 modified to restrict expression to the nucleus. Co-cultures of mKATE2-expressing target cells and PBMC effector cells (Effector: Target ratio 10:1) were seeded onto ImageLock well plates (Sartorius, Göttingen, Germany) and treated with IMC-KRAS$^{G12D}$ in the presence of NucView™ Caspase-3/7 apoptosis assay reagent (Sartorius). Images were acquired using an IncuCyte Zoom (Sartorius) equipped with a ×10 objective over 96 h at 2 h intervals. Target cell lysis, normalized to control co-cultures without IMC-KRAS$^{G12D}$ and expressed as % cytolysis, was detected using the red object metric in the IncuCyte software, and analyzed with Prism 8 software (Graphpad Software).

## CRISPR/Cas9 editing

HLA and KRAS alleles were modified using synthetic crRNA and tracrRNA were purchased from Synthego, CA (crRNA sequences: GCCTCTGTGGGGAGAAGCAA and AATGTGTGACTGCAGACCCA for HLA-A-editing; GAATATAAACTTGTGGTAGT for KRAS-editing). crRNAs were annealed to tracrRNA following manufacturer's recommendations, and the annealed cr:tracrRNAs were complexed (30 pmol each) with $3\,\mu g$ GeneArt™ Platinum™ Cas9 Nuclease (Thermo Fisher Scientific) to form ribonucleoproteins (RNP). Cell lines were transfected with the corresponding RNPs and $1\,\mu g$ linearized homology directed repair (HDR) template using Lipofectamine™ CRISPRMAX™ Cas9 Transfection Reagent (Thermo Fisher Scientific) in Opti-MEM Reduced Serum Medium (Gibco). KRAS-editing was performed in two steps.

First, using a pool of two HDR templates [one to mutate *KRAS* in PSN-1 and introduce a non-targetable (G/R)12 G mutation (introducing silent mutations at the guide target site)] and the second to introduce a G12D mutation (G12G repair template: GTATTAACCTTATGTGTGACATGT TCTAATATAGTCACATTTTCATTATTTTTATTATAAGGCCTGCTGAAA ATGACTGAGTACAAGCTTGTCGTCGTGGGCGCCGGTGGCGTAGGCAA GAGTGCGTTGACGATACAGCTAATTCAGAATCATTTTGTGGACGAAT ATGATCCAACAATAGAGGTAAATCTTGTTTTAAT and G12D repair template: GTATTAACCTTATGTGTGACATGTTCTAATATAGTCACATT TTCATTATTTTTATTATAAGGCCTGCTGAAAATGACTGAGTACAAGCT TGTCGTCGTGGGCGCCGATGGCGTAGGCAAGAGTGCGTTGACGATAC AGCTAATTCAGAATCATTTTGTGGACGAATATGATCCAACAATAGAGG TAAATCTTGTTTTAAT). The first step resulted in the isolation of 3 PSN-1 clones with KRAS^(R/G)12G and the second step transfecting a pool of the PSN-1 KRAS^(R/G)12G clones with the KRAS guide and G12D repair template resulted in 2 clones that were validated by western blot and gDNA sequencing.

*HLA-A* allele conversion was achieved using an HDR template consisting of two HLA-A*24 homology arms flanking a HLA-A*11 core region that would transcribe and translate a functional HLA-A*11 protein. The left and right homology arms were PCR amplified from the genomic DNA of PSN-1 and covered exon 1 and exon 6 of HLA-A*24, respectively, while the DNA fragment covering exons 2–5 of HLA-A*11 core was PCR amplified from genomic DNA of CL40 cells. The HLA-A*24 homology arms and HLA-A*11 core fragments were stitched by PCR and the amplified product was used as a HDR template for the HLA-A conversion of PSN-1 clones. *HLA-A* allele conversion of selected clones was validated by HLA-typing performed by VH Bio Ltd (UK).

## Western blot analysis
Cells were lysed in RIPA buffer (Thermo Fisher Scientific) containing protease and phosphatase inhibitors (Sigma-Aldrich, MO), separated using SDS-PAGE and transferred onto nitrocellulose membranes. Antibodies were diluted as described in supplementary table S7 in TBS containing 0.1% Tween-20 and 1% milk or BSA. Bound HRP secondary antibodies were detected and quantified on western blots using a C-digit scanner and Image Studio software (Li-Cor, NE).

## Reporting summary
Further information on research design is available in the Nature Research Reporting Summary linked to this article.

## Data availability
The SPR and cell-based assay data generated in this study are provided in the Source Data file. The crystallography data generated in this study have been deposited in the RCSB protein data bank (PDB) with the accession codes 7OW3, 7OW4, 7OW5, 7OW6 and 7PB2. Starting structures, parameters and input files for the molecular dynamics simulations and MMPBSA calculations are available via https://doi.org/10.5281/zenodo.6839257 Source data are provided with this paper.

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

## Acknowledgements

We would like to thank Michelle McCully, Laure Humbert, Peter Molloy, and Ita O'Kelly for critical reading and help preparing this manuscript. We also wish to thank: Vanessa Clarke, Max Beckmann, Sarah Bailey, Nicole Mai, Tein Foong-Leong, and Peter James for discussions and supporting identification of the KRAS[G12D] specific TCR and affinity maturation; Kathryn Lamming, Alex Powlesland, and Ricardo Carreira for cell line characterisation; Hemza Ghadbane for design of CRISPR guides, repair templates and editing approach used to edit HLA and KRAS alleles, and Carmine Carpenito and JoAnn Suzich for their scientific input and insights. We would like to thank Diamond Light Source and staff of beamlines for access to beamtime (proposal mx22870). A.H. and M.v.d.K thank the Engineering and Physical Sciences Research Council [grant number EP/T517872/1] and the Biotechnology and Biological Sciences Research Council [grant number BB/M026280/1] for support. Computer simulations were conducted using the computational facilities of the Advanced Computing Research Centre of the University of Bristol.

## Author contributions

A.P., V.K., A.H., S.M., T.D., K.B., M.H., C.H., C.R. J.D., C.Co., W.Y., and A.D.W. performed experiments and data analyses. J.N.H., S.H., R.R.,

A.B.L., J.D.D., M.A., N.L., M.v.d.K, G.D.P., A.V., D.K.C., A.D.W., and C.C. designed experiments and supervised the project. V.K., A.H., M.v.d.K, D.K.C., A.D.W., and C.C. wrote the manuscript.

## Competing interests

A.P., V.K., S.M., T.D., C.H., C.R., J.D., S.H., K.B., M.H., C.C., M.A., R.R., J.D.D., N.L., A.V., D.K.C., A.D.W., and C.C. are current or former employees of, and may hold shares in Immunocore Ltd. J.N.H, W.Y., A.B.L., and G.D.P. are current or former employees of, and may hold shares in Eli Lilly and Co. M.v.d.K and A.H. declare no competing interests.
