## [Peer Review File · Nature Communications]

Therapeutic high affinity T cell receptor targeting a KRAS^{G12D} cancer neoantigenREVIEWER COMMENTS

Reviewer #1 (Remarks to the Author):

The manuscript present the identification and affinity enhanced TCR that elicit T cell responses against HLA-A11-KRAS g12D, a neoantigen that has evade so many efforts. The authors present the engineered TCR with an remarkable fold increase in selectivity over the wild type

The authors fail to described clearly the differences of HLA_A*11-KRASwt vs HLA-KRASG12D in a separate section, probably the first one with a headline; now it is buried in the "The JDIA41b1 TCR adopt a virtually identical.." describing the affinity mature TCR to both HLAs. The section should clearly state that D6 is sticking down while the counterpart WT is up. The authors present that the TCR-pHLA (please note through the need to use pHLA to be clear) poses are virtually identical although the affinity are drastically different. An analysis of the contacts (numbers and types) as well as size of buried surface should be included. And discussed. The use of the word apo pHLA is incorrect and should be corrected. Apo-enzyme refers to protein/enzymes that require binding to a cofactor/coenzyme/metal to form an active complex and it is the opposite to holoenzyme. In this case the pHLA exist and displays in absence of TCR.

The statement that JDIA41b1 TCR binds via an induced fit is not supported by the data that they are virtually identical. Moreover the fact that glycine is not observed; in that line argument line 194-196 state that the peptide in the pHLA structures are in similar conformation but in the complex with the TCR-pHLA complex there is a conformational shift. The whole paragraph is paramount for the statement of induced fit but it is far from clear and needs orderly rewriting.

The work presents data using an ImmTAC using the JDI TCR as a starting point. Although the previous work has been properly referenced this section requires an scheme for this construct. in figure 4. The description of the KD weaved in the description renders it very difficult to parse. The author show the ELISPOT data but another measurement of T Cell activation is needed to solidify the data. For example the flow cytometry that they mention as 'data not shown'. Moreover the IMC-KRAS T cell mediated activation in cell lines with KRASG12d displays 1/3 of the signal as background with WT (fig 4 a) not in the. others. It is very daunting the changes in magnitude/range of the axis all IFN release should be easily comparable

Besides the issues pointed above, the manuscript has several issues that prevent its publication in its current form.

The comparisons TCR vs TCR bispecific and TCR isolated vs TCR affinity matured are not clear. The methods are not detailed enough to ensure any sort of reproducibility. The below items need to be included, expanded and corrected.

- a. The authors mention 'data not shown', this is not conducive for a publication. To sustain the argument the MS data should be included. Also details of the SPR (see below)
- b. The authors should repeat the pdb IDs in the figures and description to match and ease the table;
- c. For example what are the buffers of refolding the TCR should be fully stated not just ('redox couple?'). Too many as described! Rigor is not included
- d. Binding affinity lists 2 types of instruments Biacore T200/Biacore 8K without specifying which experiment with which instrument or conditions
- e. The sensorgrams of SPR (point b) are not included for data presented in table 1; affinity window should be defined ; table 2 should include in the title TCRs or affinity mature TCRs

f. They report that they measure binding and the description seems to attach HLA (no peptide) to the chip. This would have been impossible since HLA would be unfolded; to do exchange of peptide would work only with a peptide of tighter affinity.

g. S4 (Affinity) needs the range of concentrations use; what is A? binding of fJDIA41b1 to wt peptide? And b the steady state of a?

The text should have a thorough discussion of multicycle vs single cycle kinetics

h. The methods described 2 different pDBs for the tcr alpha and tcr beta chain. If this is accurate it should be rationalized in methods and comment it on in results

The figures

Figure. 1E needs to get the residue numbering of peptide included
Figure 2 needs labels peptide beta, etc A figure that relates the full structure to the 3 panels In S1 is necessary to appreciate where is the peptide
S2 needs dash lines to indicate the residue that has no density

Minor corrections and typos

- We suggest using G12D and WT as a superscript to ease reading as it was done in the initial sections and then dropped
- Line 70, needs 'to be'
- Reference 62 is not complete
- Table 1 of crystallography needs to be in the main manuscript with ', ' in large numerals,
- The rms bonds of 70W3, 7PB2, 7OW5 and 7OW6 are too tight (0.002-0.0032) verify it is a typo or re-refine to release ; a more typical value ranges 0.008-0.016)
- Add granularity to B values in crystallography table; for this topic b value of hla, peptide and TCR should be separated
- Define MMPBSA
- Look and add through the manuscript space between numeral and units
- Line 398, is it pi3kak1047L or rR1047L?

Reviewer #2 (Remarks to the Author):

Poole and colleagues describe T cell receptor recognition of KRAS WT and G12D epitopes presented by HLA-A11, to include characterization of a novel TcR, generation of high affinity variants, crystallographic structures, and assessment of how mutation has enhanced binding. This is followed by specificity and functional analyses, which lead the authors to conclude they have generated a variant "capable of...redirected T cell targeting of cancer cells..."

This is a particularly exciting area to work - targeting of KRAS is indeed of considerable interest. The authors have done an extensive amount of work, and in the end likely produced a molecule worthy of further study. However, I have very significant concerns with the manuscript. Some of these are easily addressed through rewriting, but others likely require more extensive work and reconsideration of conclusions.

1) Much of the manuscript is significantly over-hyped in ways that detract from the main findings. The abstract, for example, ends with "These findings reveal a novel molecular mechanism for TcR selectivity for neoantigens that differ from self antigens by only a single amino acid..." (lines 44-46). Of course each TCR will act a little differently, but as with any biomolecular interactions there are a myriad of ways specificity can arise, and sensing conformational changes indirectly via electrostatics (as the authors propose) is hardly a "novel" mechanism. Additionally, we have known for decades that TcRs can sense single amino acid variations in peptides - seeing this happen in an interesting neoantigen is hardly novel. Sure, the exact details here make it technically novel (see point 2 below), but this is not what the authors mean. This sentence in the abstract typifies much of the manuscript, where unnecessary hype and claims of something new and interesting diminishes from what has been actually achieved.

2) Following on the point above, at the end of the introduction (lines 105-107) the authors write "This intriguing mechanism has important implications for understanding..." Each TCR - pep-MHC interaction will be different. How could what is observed here be adapted to other systems? It certainly could not be engineered in a rationale way or otherwise coopted for another system. The mechanism the authors purport to have identified does not have "obvious utility for developing cancer therapies that can target this pathway" as the authors write. Maybe the experiments performed do - generating a high affinity TCR, characterizing, making a bispecific do. But the mechanism uncovered? See point 1 about hype detracting from the message.

3) Results in Figure 1 - what are the concentrations used in Fig. 1A? How many times were these experiments repeated to generate the SEM error bars? For the affinity measurements in B, C, and D - where are the errors? How many times were the SPR experiments repeated? These questions

of errors and indications of replicates and reproducibility are a problem throughout the paper. There are no errors on the affinity measurements anywhere, no errors on the thermodynamics, etc.

4) On line 130 in the results, the authors describe the efforts to generate a higher affinity TcR, opening with the point that "higher affinity enables effective targeting of antigens..." There is no acknowledgement here or anywhere else of the risk of enhancing cross-recognition of other antigens. It is implicitly acknowledged via specificity analyses later on, but to not address a point the entire field has been wary of for nearly 10 years (since high affinity TcRs killed people!) is highly unusual. This is a major problem throughout the paper.

5) Lines 145-146 - "...without apparent loss of specificity." This point seems to be solely based on G12D-WT comparisons -- so at this point in the paper, it is meaningless as specificity is well understood to take into account more than cross-recognition of two peptides (see point 1). This is particularly the case because of the well acknowledged danger and skepticism about high affinity TcRs and their risks about specificity, a point which the authors ignore throughout (see point X below).

6) We need more detail about the library screen described on lines 148-153. It is not clear from the methods how big this library was, is it clear what the 1182 peptides actually represent. This is a crucial part of the story and the authors sum it up in one panel in Fig. 1. Given the prior results with high affinity TcRs, this screen, its description and associated presentation of results warrants its own figure and an extended discussion.

7) Following the point above, it is also not clear WHICH receptor was used in this screen, and what its affinity was. Why did the authors not do all of these? We need to see them all, as we might expect different results as a function of affinity.

8) The authors should validate their specificity screens with sub libraries of defined peptides, ideally with transduced cells.

9) For the structures, the authors need to show 2Fo-Fc density from composite omit maps with associated discussion about limitations of interpretations. This should be done for peptides as well as key regions of the TcRs.

10) The greater energetic contribution for binding of the mutant is of interest given the GD mutation. However, I found the molecular dynamics decomposition of binding energies unconvincing. They add little to the story and the agreement in overall energies is likely fortuitous. To validate these, the authors need to make mutations, measure energetics, perform similar simulations with mutated proteins, and show agreement between experiment and calculation. The overall conclusion from this section (lines 284-292) that the energetic changes are a consequences of the GD mutation is obvious without the simulations and associated discussion, and supported by the thermodynamic measurements (a greater use of the simulation here would be exploring how the mutation impacts the propensity for the peptide to undergo its conformational change).

11) Following point 10, what electrostatic surface potentials are being shown in Fig. 3E and F? Are these from static structures?

12) Lines 294 - 299 - it is not clear which affinity enhanced TcR is being tested here. Is it and its sequence shown in Table 2? Is it one of the ones whose specificity was examined via the library? (see point 7).

13) Figure 4 - again, how many replicates/repeats?

14) Line 330-337 - this data must be shown. It speaks to the potential for off target recognition (albeit using other HLAs) by the high affinity TcR. Other places where data not shown is indicated needs to show the data (line 347).

15) Discussion: line 385 - 386 needs attention. The challenge is not one for us, but one for the immune system. This clearly can and does happen. That the authors have managed to engineer a receptor that maintains this degree of selectivity is nice, but this does not need to be sold as something that is a high challenge, as the immune system already does this (see point 1).

16) Line 402 - 405: while an achievement, others have done the same thing with TcRs in other systems, although not with a neoantigen. The word "Remarkably" is unnecessary here.

17) To the point above, the authors have been selective in their citations. There have been many studies of TcR engineering, assessments of specificity, conformational changes, etc. The authors can and should be more inclusive.

18) Line 416-418: this conclusion is clear without the molecular dynamics simulations.

19) Line 428-430: TcRs have been shown to use a variety of complex mechanisms to achieve selectivity towards different antigens. Moreover, there are no studies here of peptide dynamics as the authors imply (although that could be interesting as written above).

20) Lines 446-447: TcRs have been sensing single amino acid changes in peptides for as long as we have been studying them. The fact that T cells can target neoantigens already implies this process can happen with neoantigens without any of the data here. See point 1.

21) A major point: I found it striking that the authors did not mention at all the danger of high affinity TcRs resulting from the risks of off-target toxicity. They address this somewhat with their specificity screens and the functional studies with the bispecific, but the study is not particularly comprehensive in screening. That's ok, but do not even address this point is quite striking.

22) Did I miss the reference in the paper to the structure with the WT TcR? Why is there no comparison of the high affinity version to this? It would seem relevant to points about specificity, etc.

Reviewer #3 (Remarks to the Author):

Poole and colleagues describe and functionally characterize a novel TCR-anti-CD3 scFv fusion protein directed against a KRASG12D peptide presented by HLA-A*11. The results are well presented and the conclusions are supported by the data provided. Based on the high frequency of the KRAS-G12D mutation in many solid tumors, this study has a high translational potential. However additional experiments must be performed to demonstrate the in vivo anti-tumor function of high affinity T cell receptor targeting the KRASG12D cancer neoantigen.

A disadvantage of the approach is the HLA-A*11 restriction of the ImmTAC. Even though HLA-A*11 is one of the more frequent HLA-A alleles, it is only expressed in around 5-15% of the Europeans/North Americans and 20-30% of the East Asian populations. Thus this therapy would be only be available for a minority of the patients with KRASG12D mutated tumors.

Specific comments:

Fig. 1A-D, Fig. 4A-G: numbers (n) of repeated experiments should be indicated, error bars are missing in Fig. 1B.

Fig. 1A: The figure legend states that the non-transduced controls are not shown, even though they appear in the graph.

Fig. 1B: According to the figure legend and axis description, bars in Fig. 1B represent the fold change of KD (dissociation constant) of JDI TCR towards mutated peptides relative to G12D. Consequently, when the affinity to the mutated peptide is lower than the affinity to G12D, the ratio is superior to 1. When the affinity to the mutated peptide is higher than the affinity to G12D, the ratio is inferior to 1. Most of the bars indicate values inferior to 1 and, thus, most of the mutated peptides display a higher affinity to JDI TCR than G12D. Therefore, I don't understand the author's statement in the text saying that peptide residue D6, G4, A5 and G9 are most critical for JDI TCR binding. In addition, I would have expected the affinity of G12D to JDI TCR to decrease when

individual amino acids of the peptide are mutated.

Fig. 1 and Table 2: In the text Z142, the "affinity window" is defined as fold change in affinity of JDI TCR to KRASG12D-HLA-A*11 compared to KRASWT-HLA-A*11. Although, the figure legend Fig. 1D defines the "affinity window" as fold change of the affinity of JDI TCR relative to parental (non-affinity matured?) TCR. In both cases the same values of the "affinity window" are shown.

Table 2: The "affinity window" of JDIa9bwt in Table 2 and of the JDI TCR in Z298 was apparently calculated as $KD\ KRASWT-HLA-A*11 / KD\ KRASG12D-HLA-A*11$. However, when applying the same calculation to JDI a41b1 in Table 2 the "affinity window" should be much higher than indicated. Please indicate a clear and consistent definition of the "affinity window".

Fig. 1/Table 2: Within the text the authors mention $t_{1/2}$ measurements of affinity enhanced TCRs. Please indicate the values, f.e. in Table 2.

Fig. 2A,C,E; Fig. 3A: Superposed graphical representation of two/four (in Fig. 4E) peptides within one image are very difficult to understand. Thus, when structural differences of peptides should be highlighted and are important for the message, I would rather represent them in two images side by side than overlaid. I.e., differences are much easier to see in Fig. 3E,F where two juxtaposed images show the differences.

Fig. 4: Figure legends 4A and B are interchanged

Fig. 4D: Please show relative intensities of KRASG12D/GAPDH bands for different clones to allow the reader to compare expression levels with IFN γ response after co-culture with PBMCs in Fig. 4E.

Fig. 4G: Were other markers than IFN γ increased such as perforin, granzyme, CD107a etc.?

Fig 4F: The in vitro cytolytic activity is impressive but in order to understand the translational potential in vivo experiments in the mouse model using KRAS mut cancer cell lines or genetic models are required.

KRAS-G12D was recently shown to induce NLRP3 activation (Nat Commun. 2020 Apr 3;11(1):1659. doi: 10.1038/s41467-020-15497-1) – how would that affect the anti-KRA-G12D T cell response given that continuous IL-1 β exposure may cause T cell exhaustion?

Comments on Methods:

Please describe in more detail how the KRASG12D-HLA-A*11-directed TCR was identified and how cells were pulsed with peptides.

Manuscript number: NCOMMS-21-35973; submission date 9th Sept 2021

Reference expertise: TCR structure/Neoantigens/structural biology/Immune therapy in KRAS driven cancer

RESPONSE TO REVIEWER COMMENTS

Reviewer #1:

The manuscript present the identification and affinity enhanced TCR that elicit T cell responses against HLA-A11-KRAS g12D, a neoantigen that has evade so many efforts. The authors present the engineered TCR with an remarkable fold increase in selectivity over the wild type

R1_General response: We thank the reviewer for appreciating our work on engineering a TCR against a neoantigen and achieving a remarkable fold increase in selectivity. We are very thankful for the constructive comments and have respectfully tried to address them:

*R1_Comment 1: The authors fail to described clearly the differences of HLA_A*11-KRAS^{wt} vs HLA-KRAS^{G12D} in a separate section, probably the first one with a headline; now it is buried in the “The JDIa41b1 TCR adopt a virtually identical..’ describing the affinity mature TCR to both HLAs. The section should clearly state that D6 is sticking down while the counterpart WT is up.*

R1_Author response 1: We thank the reviewer for this suggestion. We have modified the manuscript to include a separate section titled “KRAS^{G12D} peptide D6 residue acts as a secondary HLA anchor in the HLA-A*11-KRAS^{G12D}-TCR complex” (**Line 187-196**) to clearly explain the differences in HLA-KRAS^{WT} and HLA-KRAS^{G12D} peptide conformations in TCR bound structures. However, we respectfully disagree with the reviewer’s statement that “D6 is sticking down while the counterpart WT is up”. This conclusion is not made in the manuscript or supported by the data.

R1_Comment 2: The authors present that the TCR-pHLA (please note through the need to use pHLA to be clear) poses are virtually identical although the affinity are drastically different. An analysis of the contacts (numbers and types) as well as size of buried surface should be included. And discussed.

R1_Author response 2: A new supplementary table (**Table S2**) has been added with the number and type of contacts, and buried surface area data for the peptide and HLA for the JDIa41b1 TCR

R1_Comment 3: The use of the word apo pHLA is incorrect and should be corrected. Apo-enzyme refers to protein/enzymes that require binding to a cofactor/coenzyme/metal to form an active complex and it is the opposite to holoenzyme. In this case the pHLA exist and displays in absence of TCR.

R1_Author response 3: We thank the reviewer for highlighting this point. We agree that the term apo- is inappropriate to use in the context of peptide-HLA complex. We have modified

the manuscript to remove the term from the text (**Lines 228-229. 490, 1107-1109**) and **Fig 2e** to clarify that structures are pHLA structures without a bound TCR, for example:

Lines 229-230: ‘structures of both pHLAs without a bound TCR’

R1_Comment 4: The statement that JDIa41b1 TCR binds via an induced fit is not supported by the data that they are virtually identical. Moreover the fact that glycine is not observed; in that line argument line 194-196 state that the peptide in the pHLA structures are in similar conformation but in the complex with the TCR-pHLA complex there is a conformational shift. The whole paragraph is paramount for the statement of induced fit but it is far from clear and needs orderly rewriting.

R1_Author response 4: We understand the reviewer’s perspective but respectfully disagree with the comment that the data doesn’t support the statement that JDIa41b1 TCR binds via an induced fit.

Both peptides HLA-KRAS^{WT} and HLA-KRAS^{G12D} adopt a similar but bulged conformation in the absence of TCR (Fig 2e). But in the TCR bound form, both peptides undergo a conformational shift to adopt a buried conformation (Fig 2f). The switch from bulged conformation to buried conformation is induced by TCR.

We have revised the text in the first paragraph (beginning **line 226**) for clarity and hope that the subsequent paragraph explains clearly the induced conformational shift in the TCR bound structures (**line 239-247**)

R1_Comment 5: The work presents data using an ImmTAC using the JDI TCR as a starting point. Although the previous work has been properly referenced this section requires an scheme for this construct. in figure 4. The description of the KD weaved in the description renders it very difficult to parse.

R1_Author response 5: We thank the reviewer for the suggestion to improve the description of the ImmTAC molecule and have now included a schematic diagram (**Fig 1e**) of the steps and enhancement in K_D from JDI TCR to IMC-KRAS^{G12D} ImmTAC

R1_Comment 6: The author show the ELISPOT data but another measurement of T Cell activation is needed to solidify the data. For example the flow cytometry that they mention as ‘data not shown’.

R1_Author response6: We thank the reviewer for this suggestion. Additional data demonstrating T cell activation by detection of CD69 and CD25 upregulation in T cells co-cultured with target cells presenting KRAS^{G12D} peptide and incubated with IMC-KRAS^{G12D} have now been included in the supplementary data (**Supplementary Figure S8**) to support the ELIPSOT data presented.

R1_Comment7: Moreover the IMC-KRAS T cell mediated activation in cell lines with KRASG12d displays 1/3 of the signal as background with WT (fig 4 a) not in the. others. It is very daunting the changes in magnitude/range of the axis all IFN release should be easily comparable

R1_Author response7: We have revised the axis of the IFN γ release assay plots shown in **Fig 5a, b and e** to make the scale of the axis consistent. The purpose of these dose response experiments is to establish a dose-effect relationship for the TCR-CD3 bispecific protein. Importantly, we would like to highlight to the reviewer that the IFN γ signal mediated by IMC-KRAS^{G12D} incubated with target cells expressing KRAS^{WT} is both lower in magnitude, but also occurs at concentrations of IMC-KRAS^{G12D} 1-2 Log greater than concentrations of IMC-KRAS^{G12D} that mediate specific T cell activation to cells presenting KRAS^{G12D} pHLA, creating the potential for a therapeutic window.

R1_Comment8: Besides the issues pointed above, the manuscript has several issues that prevent its publication in its current form.

The comparisons TCR vs TCR bispecific and TCR isolated vs TCR affinity matured are not clear.

R1_Author response 8: We refer the reviewer to our response to comment 5. We have added a schematic showing the TCR, affinity matured TCR and TCR bispecific (**Fig 1e**). In addition, we have amended **Table 2** to clearly compare different versions of the TCR.

R1_Comment9: The methods are not detailed enough to ensure any sort of reproducibility. The below items need to be included, expanded and corrected.

a. The authors mention 'data not shown', this is not conducive for a publication. To sustain the argument the MS data should be included. Also details of the SPR (see below)

R1_Author Response 9a: We agree with the reviewer's comment and have either included the data that was otherwise referred to as 'data not shown' or removed reference to it where we felt that the data did not meaningfully add to the manuscript and the data presented.

b. The authors should repeat the pdb IDs in the figures and description to match and ease the table;

R1_Author Response 9b: PDB IDs are now included in the figure legends of **Fig 2, S2-5** in the revised manuscript

c. For example what are the buffers of refolding the TCR should be fully stated not just ('redox couple?'). Too many as described! Rigor is not included

R1_Author Response 9c: Detailed methods have been described in the previous publications that we referenced and therefore we felt it may not be necessary to describe it again. However, we are happy to accept the reviewer's suggestion and have included further detail, for example of buffers used for TCR refolding in the materials and methods section

Lines 599-604: 'TCR α and TCR β chain inclusion bodies were denatured using 50 mM Tris buffer pH 8.1 containing 100mM Sodium Chloride, 6 M Guanidine and 20 mM Dithiothreitol (DTT) and refolded by diluting to final protein concentration of 60 mg/L into a buffer containing 100 mM Tris pH 8.1, 4 M Urea, 400 mM L-Arginine, 1.9 mM Cystamine and 6.5 mM Cysteamine redox couple. Refolding mixture was dialysed against water followed by 20 mM Tris buffer pH8.1.'

d. Binding affinity lists 2 types of instruments Biacore T200/Biacore 8K without specifying which experiment with which instrument or conditions

R1_Author Response 9d: We thank the reviewer for highlighting this. We have revised the manuscript to include details on usage of Biacore instruments and conditions used for affinity measurements in the methods section (**Lines 616-625**).

e. The sensorgrams of SPR (point b) are not included for data presented in table 1; affinity window should be defined ; table 2 should include in the title TCRs or affinity mature TCRs

R1_Author Response 9e: We thank the reviewer for highlighting these omissions. Sensorgrams of SPR measurements for the TCRs reported in **Table 2** have been included in the supplementary data (**Fig S1**) and the titles of the TCR indicate where the TCR is affinity enhanced. We have revised the manuscript to define the affinity window in the text on line 144-5, in the legend for Figure 1D (**line 1086**) and in the column title of **Table 2**.

Line 144-5: ‘widening the affinity window (K_D KRAS^{WT}/ K_D KRAS^{G12D} pHLA),’

f. They report that they measure binding and the description seems to attach HLA (no peptide) to the chip. This would have been impossible since HLA would be unfolded; to do exchange of peptide would work only with a peptide of tighter affinity.

R1_Author Response 9f: For clarity, in this study peptide-HLA complexes (pHLA) were immobilised on the SPR sensor chip. It is a stable complex suitable for measurement of TCR affinity. We have now revised the materials and methods in the manuscript clearly stating that pHLA was used in these experiments. .

Line 609: Peptide-HLA (pHLA) complexes were refolded

g. S4 (Affinity) needs the range of concentrations use; what is A? binding of fJDIa41b1 to wt peptide? And b the steady state of a?

The text should have a thorough discussion of multicycle vs single cycle kinetics

R1_Author Response 9g: We have revised the manuscript in response to the reviewer’s comment and now include sensorgrams of SPR measurements for the JDI TCR and high affinity mutants JDIa41b1 and IMC-KRAS^{G12D} in **Supplementary Figure 1**. Further experimental details have been provided in the materials and methods section (**line 617-627**) and in the legend of Supplementary Figure 1.

h. The methods described 2 different pdbs for the tcr alpha and tcr beta chain. If this is accurate it should be rationalized in methods and comment it on in results

R1_Author Response 9h: We have revised the materials and methods section of the manuscript in response to the reviewers comment :

Line 654-660: ‘Structures were solved by molecular replacement using using PDB 4UQ2 for HLA and B2m, and a hybrid TCR generated by combining PDB 4JRX (TCR α chain) and PDB 4X6B (TCR β chain) as search models in Phaser⁵³. Two separate PDBs were chosen for the TCR α and β chains as these individually have very high sequence identity (> 90%) for the JDI TCR chains. The

TCR β chain from PDB 4X6B was superposed onto and in the place of CA5 TCR beta chain (PDB 4JRX) to create the hybrid TCR search model'

The figures

Figure. 1E needs to get the residue numbering of peptide included

R1_Author Response 10a: We thank the reviewer for this suggestion. The peptide information previously shown in Fig 1e is now presented later in the manuscript, in **Table 5** where the peptide residues are numbered for clarity.

Figure 2 needs labels peptide beta, etc A figure that relates the full structure to the 3 panels In S1 is necessary to appreciate where is the peptide

S2 needs dash lines to indicate the residue that has no density

R1_Author Response 10b: We thank the reviewer for these suggestions and agree. **Figures 2, S1 and S3** have been updated in line with the reviewer's suggestions.

Minor corrections and typos

- We suggest using *G12D* and *WT* as a superscript to ease reading as it was done in the initial sections and then dropped

- Line 70, needs 'to be'

- Reference 62 is not complete

- Table 1 of crystallography needs to be in the main manuscript with ', ' in large numerals,

- The rms bonds of 70W3, 7PB2, 7OW5 and 7OW6 are too tight (0.002-0.0032) verify it is a typo or re-refine to release ; a more typical value ranges 0.008-0.016)

- Add granularity to B values in crystallography table; for this topic b value of hla, peptide and TCR should be separated

- Define MMPBSA

- Look and add through the manuscript space between numeral and units

- Line 398, is it pi3kak1047L or rR1047L?

R1_Author Response 10c: We apologise for these typographical errors and thank the reviewer for bringing them to our attention. The errors have been corrected and the edits marked up by yellow highlighting in the manuscript. The structures have been re-refined to release the tight restraints (**Table 3**). In addition, **Table 3** has been updated with separate B value rows for HLA, peptide and TCR chains

Reviewer #2:

Poole and colleagues describe T cell receptor recognition of KRAS WT and G12D epitopes presented by HLA-A11, to include characterization of a novel TcR, generation of high affinity variants, crystallographic structures, and assessment of how mutation has enhanced binding. This is followed by specificity and functional analyses, which lead the authors to conclude they have generated a variant "capable of...redirected T cell targeting of cancer cells..." This is a particularly exciting area to work - targeting of KRAS is indeed of considerable

interest. The authors have done an extensive amount of work, and in the end likely produced a molecule worthy of further study. However, I have very significant concerns with the manuscript. Some of these are easily addressed through rewriting, but others likely require more extensive work and reconsideration of conclusions.

R2_General response: We thank the reviewer for appreciating our work engineering a soluble TCR bispecific protein targeting KRAS^{G12D} and agree that this is an exciting field. We are very thankful for the constructive comments and have tried to respectfully address the specific comments:

R2_Comment1: Much of the manuscript is significantly over-hyped in ways that detract from the main findings. The abstract, for example, ends with "These findings reveal a novel molecular mechanism for TcR selectivity for neoantigens that differ from self antigens by only a single amino acid..." (lines 44-46). Of course each TCR will act a little differently, but as with any biomolecular interactions there are a myriad of ways specificity can arise, and sensing conformational changes indirectly via electrostatics (as the authors propose) is hardly a "novel" mechanism. Additionally, we have known for decades that TcRs can sense single amino acid variations in peptides - seeing this happen in an interesting neoantigen is hardly novel. Sure, the exact details here make it technically novel (see point 2 below), but this is not what the authors mean. This sentence in the abstract typifies much of the manuscript, where unnecessary hype and claims of something new and interesting diminishes from what has been actually achieved.

R2_Author response 1: We thank the reviewer for the comment and structured argument about biomolecule specificity. We agree that TCRs will have very many ways to achieve specificity with small differences in binding interactions or electrostatic interactions. What we find intriguing is that indirect electrostatic interactions alone can drive a large affinity window (>4000 fold) without any differences in direct contacts, and this forms the basis of specificity for a particular high affinity TCR against a neoantigen. It was not our intention to convey hype or exaggerate novelty. We respect the reviewer's opinion and have addressed this point, rephrasing the abstract and discussion for example

Lines 44-47: 'These findings reveal an intriguing molecular mechanism driving the exquisite selectivity of a high affinity TCR bispecific molecule against a shared neoantigen target with therapeutic potential.'

R2_Comment2: Following on the point above, at the end of the introduction (lines 105-107) the authors write "This intriguing mechanism has important implications for understanding...." Each TCR - pep-MHC interaction will be different. How could what is observed here be adapted to other systems? It certainly could not be engineered in a rationale way or otherwise coopted for another system. The mechanism the authors purport to have identified does not have "obvious utility for developing cancer therapies that can target this pathway" as the authors write. Maybe the experiments performed do - generating a high affinity TCR, characterizing, making a bispecific do. But the mechanism uncovered? See point 1 about hype detracting from the message.

R2_Author response 2: We thank the reviewer for this feedback and understand the opinion that the conclusions need to focus on the novelty and value of the methods and reagents generated in this work, whereas the mechanism adds to the growing body of evidence that

supports understanding of TCR-pHLA interactions to achieve specificity. We have rephrased the sentence at the end of the introduction section so that it reads:

Lines 101-106: “Using this affinity enhanced TCR as the targeting arm, a bispecific T cell engaging ImmTAC (Immune mobilizing monoclonal TCR Against Cancer) molecule²¹, mediated selective T cell targeting of HLA-A*11+ cancer cells naturally expressing KRAS^{G12D}. These findings highlight the exquisite sensitivity of the TCR:pHLA system and imply that soluble high affinity TCR bispecifics may have the potential to treat neoantigen driven cancers.”

R2_Comment3: Results in Figure 1 - what are the concentrations used in Fig. 1A? How many times were these experiments repeated to generate the SEM error bars? For the affinity measurements in B, C, and D - where are the errors? How many times were the SPR experiments repeated? These questions of errors and indications of replicates and reproducibility are a problem throughout the paper. There are no errors on the affinity measurements anywhere, no errors on the thermodynamics, etc.

R2_Author response 3: We would like to thank the reviewer for bringing the figure legend text to our attention and for the suggestion to provide indications of replicates and reproducibility in the data. We have modified the Figure 1 legend to clearly state the concentration of peptide used in **Fig 1a (line 1076)**. We have also modified the bar charts shown to indicate replicate data points and have clarified the text in the fig legends on **pages 35-43**. For clarity, the majority of the experiments shown in the figures are representative examples of experiments that were repeated on at least two, often more, independent occasions to ensure the results were robust and reproducible. Each individual experiment included duplicate or triplicate samples treated identically.

The affinity data for most individual mutants (indicated by grey) in Fig 1c and d were single measurements. However, the affinity measurements for the mutants investigated in more detail in the manuscript (indicated by black data points in Fig 1c & d), for example JDI TCR, JDIa9bwt, JDIa41b1 and JDIa96b35 / IMC-KRAS^{G12D} were measured in two independent experiments and error bars have now been included on all the affinity measurements (**Fig1b, Fig S1**) and thermodynamics plots (**Fig S6**).

R2_Comment4: On line 130 in the results, the authors describe the efforts to generate a higher affinity TcR, opening with the point that "higher affinity enables effective targeting of antigens..." There is no acknowledgement here or anywhere else of the risk of enhancing cross-recognition of other antigens. It is implicitly acknowledged via specificity analyses later on, but to not address a point the entire field has been wary of for nearly 10 years (since high affinity TcRs killed people!) is highly unusual. This is a major problem throughout the paper.

R2_Author response 4: We thank the reviewer for this feedback and understand that there are examples in the literature where TCR affinity enhancement has led to cross-reactivity. We have revised the manuscript, which no longer contains the text in the above-mentioned line 130, adding new data and text addressing the potential for cross-reactivity of the high affinity TCR starting at **line 330** (see also our response to reviewer comments 5, 6, 7, 8).

We acknowledge the historic and unfortunate cases where a high affinity TCR in the context of an adoptive T cell therapy (directed to MAGE-A3 pHLA) lead to patient mortality. We

have included a reference to this case in the discussion section on line 514 and furthermore, we additional data using methods capable of detecting the cross-reactivity associated with the problematic MAGE-A3 TCR including cell based assays using cardiac cell lots (**Figure 5e**) and a molecular and functional analysis of specificity (**Figure 4** and **Table 5**)

R2_Comment5: Lines 145-146 - "...without apparent loss of specificity." This point seems to be solely based on G12D-WT comparisons -- so at this point in the paper, it is meaningless as specificity is well understood to take into account more than cross-recognition of two peptides (see point 1). This is particularly the case because of the well acknowledged danger and skepticism about high affinity TcRs and their risks about specificity, a point which the authors ignore throughout (see point X below).

R2_Author response 5: We have clarified the text to make clear that at this point in the manuscript the focus of the specificity assessment is KRAS^{WT} given the peptide sequence homology:

Line 148-150: ‘one million-fold affinity enhancement over the parent JDI TCR with enhanced selectivity relative to the ubiquitously expressed HLA-A*11-KRAS^{WT}.’

We refer the reviewer to our response to comments 4 and 6. We have revised the manuscript to include additional, more detailed assessment of the TCR specificity later in the manuscript, for example **Figure 4**, **Figure 5e** and **Table 5**.

R2_Comment6: We need more detail about the library screen described on lines 148-153. It is not clear from the methods how big this library was, is it clear what the 1182 peptides actually represent. This is a crucial part of the story and the authors sum it up in one panel in Fig. 1. Given the prior results with high affinity TcRs, this screen, its description and associated presentation of results warrants its own figure and an extended discussion.

R2_Author response 6: We thank the reviewer for this comment and agree that this data warranted more description and discussion. We have added a separate section in the results focussing on the library screen titled “Affinity enhanced JD1a96b35 TCR binds KRAS^{G12D}-pHLA with high specificity” (Beginning at **line 330**), as well as a dedicated Figure (**4**) presenting additional biochemical and cell-based assay data interrogating the specificity of the molecule. We have also provided details of the library in the methods section:

Line 691-692: ‘with an estimated library size of 6.6×10^9 colonies⁶¹.’

R2_Comment7: Following the point above, it is also not clear WHICH receptor was used in this screen, and what its affinity was. Why did the authors not do all of these? We need to see them all, as we might expect different results as a function of affinity.

R2_Author response 7: We apologise if it was unclear which TCR was used in specific experiments. For clarity the TCR JD1a96b35 incorporated into the bi-specific molecule IMC-KRAS^{G12D} was used to screen the scHLA library. We have amended the manuscript to clearly state this as well as the binding affinity of the molecule:

Line 350-351: ‘This panel of peptides was then used to assess JD1a96b35 TCR binding to peptide-HLA-A*11 complexes other than HLA-A*11-KRAS^{G12D} using SPR’

We believe that this is the most relevant molecule to assess specificity. Furthermore, the alanine scan of the JDI parent TCR shown in Fig 1b is consistent with the core binding motif shown in Fig 4 despite a large improvement in affinity.

R2_Comment8: The authors should validate their specificity screens with sub libraries of defined peptides, ideally with transduced cells.

R2_Author response 8: We thank the reviewer for this suggestion and refer to our response to comment 6 - a separate section describing the pHLA library screen has been added (at **Line 330**), including additional data presented in **Fig 4** validating the screen using cell assays and panels of peptides.

R2_Comment9: For the structures, the authors need to show 2Fo-Fc density from composite omit maps with associated discussion about limitations of interpretations. This should be done for peptides as well as key regions of the TCRs.

R2_Author response 9: In agreement with the reviewer's comment, **new supplementary figure S7** has been added to show 2Fo-Fc density from composite omit maps for the peptides and CDR regions for JDIa41b1 TCR. Interpretations from the density have also been added to the results.

R2_Comment10: The greater energetic contribution for binding of the mutant is of interest given the GD mutation. However, I found the molecular dynamics decomposition of binding energies unconvincing. They add little to the story and the agreement in overall energies is likely fortuitous. To validate these, the authors need to make mutations, measure energetics, perform similar simulations with mutated proteins, and show agreement between experiment and calculation. The overall conclusion from this section (lines 284-292) that the energetic changes are a consequences of the GD mutation is obvious without the simulations and associated discussion, and supported by the thermodynamic measurements (a greater use of the simulation here would be exploring how the mutation impacts the propensity for the peptide to undergo its conformational change).

R2_Author response 10: We are sorry that the reviewer deemed the decomposition of binding energies unconvincing. We would like to point out that this approach is using physically realistic simulations to pinpoint the causes of the difference in binding energy, rather than making assumptions based on the nature of the mutation (or static structures). We thus believe this analysis is of value.

We respectfully disagree with the statement that “the agreement in overall energies is likely fortuitous”. We have previously used similar protocols based on MD simulation and MM/PBSA binding affinity analysis for other TCR-pHLA complexes, and consistently found agreement between experimental and calculated binding energy differences when comparing wild-type TCRs and their affinity enhanced variants (see: <https://doi.org/10.1016/j.omto.2020.07.008>). The exact details of the simulation protocols used here were chosen based on a careful application and comparison of various options applied to TCR-pHLA complexes (<https://www.biorxiv.org/content/10.1101/2021.06.21.449221v1>). We could therefore expect at least reasonably good agreement in overall energies.

That the difference in affinity and energetic changes are due to the GD mutation is indeed obvious, as the reviewer suggests, because this is the only difference between the two systems. However, as indicated above, the point of the detailed simulation analysis is to gain insight into the mechanistic reason for this difference. Without running the MD simulation and subsequent analysis, we would not have known how much of the difference in affinity is directly due to the electrostatic effects of the GD mutation, or (for example) knock-on effects due to the difference in peptide structure or flexibility.

R2_Comment11: Following point 10, what electrostatic surface potentials are being shown in Fig. 3E and F? Are these from static structures?

R2_Author response 11: The potentials shown are indeed calculated using static structures, after initial energy minimisation (prior to MD simulation). We have now clarified this in the figure 3 legend as follows:

Line 1131-1132: Based on static structures prepared for simulation. The scale used is $\pm 2eV$.

R2_Comment12: Lines 294 - 299 - it is not clear which affinity enhanced TcR is being tested here. Is it and its sequence shown in Table 2? Is it one of the ones whose specificity was examined via the library? (see point 7).

R2_Author response 12: We apologise again if it was unclear which TCR was used in specific experiments and refer the reviewer to our response to comment 7. We have revised the manuscript to clarify that the TCR used for scHLA library screening is the same as the one developed into ImmTAC bispecific for further assessment in cellular screening.

R2_Comment13: Figure 4 - again, how many replicates/repeats?

R2_Author response 13: We again thank the reviewer for bringing the figure legend text to our attention and corrected the text as referred to in our response to point 3.

R2_Comment14: Line 330-337 - this data must be shown. It speaks to the potential for off target recognition (albeit using other HLAs) by the high affinity TcR. Other places where data not shown is indicated needs to show the data (line 347).

R2_Author response 14: We agree with the reviewer's comment and have either included the data that was otherwise referred to as 'data not shown' or removed reference to it where we felt that the data did not meaningfully add to the manuscript and the data presented.

R2_Comment15: Discussion: line 385 - 386 needs attention. The challenge is not one for us, but one for the immune system. This clearly can and does happen. That the authors have managed to engineer a receptor that maintains this degree of selectivity is nice, but this does not need to be sold as something that is a high challenge, as the immune system already does this (see point 1).

R2_Author response 15: We agree with the reviewer that the initial challenge is one for the immune system i.e. to generate a TCR against neoantigen. However, we respectfully disagree with the reviewer that it is not also challenging to affinity enhance a TCR whilst retaining high selectivity. Indeed, we refer the reviewer to their earlier comment 4 describing instances

where engineered high affinity TCRs used in the context of adoptive cell therapies have resulted in cross-reactivity. Generating a high affinity TCR with a high degree of selectivity is an appreciable effort which involves iterative engineering and thorough screening. We have however, removed the sentence referred to in the reviewer's comment as on reflection, it was no longer necessary for the discussion.

R2_Comment16: Line 402 - 405: while an achievement, others have done the same thing with TcRs in other systems, although not with a neoantigen. The word "Remarkably" is unnecessary here.

R2_Author response 16: We agree with the reviewer and have removed the word remarkably from the text in the revised manuscript:

Line 484: '...and it was possible to enhance the binding affinity of this TCR to HLA-A*11-KRAS^{G12D} by over a million-fold, while...'

R2_Comment17: To the point above, the authors have been selective in their citations. There have been many studies of TcR engineering, assessments of specificity, conformational changes, etc. The authors can and should be more inclusive.

R2_Author response 17: We thank the reviewer for the suggestion. We agree there have been many studies in TCR engineering and specificity assessment. We believe we have included as many articles (**References 35-43**) as we found relevant to the data presented in the manuscript.

R2_Comment18: Line 416-418: this conclusion is clear without the molecular dynamics simulations.

R2_Author response 18: As indicated in our response to reviewer comment 10, the *hypothesis* of the increased negative electrostatics on the HLA surface (due to the GD mutation), increasing the electrostatic interactions with the TCR, could indeed have been made without the simulations. There may have been alternative explanations, however, with electrostatic screening of the buried D residue possibly diminishing the direct electrostatic effect. Through the (physically realistic) MD simulations and the subsequent binding decomposition analysis, we provide strong support for our hypothesis. The conclusion could not have been made confidently in the absence of the simulations.

R2_Comment19: Line 428-430: TcRs have been shown to use a variety of complex mechanisms to achieve selectivity towards different antigens. Moreover, there are no studies here of peptide dynamics as the authors imply (although that could be interesting as written above).

R2_Author response 19: We agree with the reviewer that additional experiments on peptide dynamics could have been interesting. Given the scope of the paper we have focussed our MD simulations on understanding how TCR interactions differ between the wildtype and mutated peptide. We would also like to draw attention to our responses to comments 10 and 18.

R2_Comment20: Lines 446-447: TcRs have been sensing single amino acid changes in peptides for as long as we have been studying them. The fact that T cells can target neoantigens already implies this process can happen with neoantigens without any of the data here. See point 1.

R2_Author response 20: We refer the reviewer to our response to comment 1.

R2_Comment21: A major point: I found it striking that the authors did not mention at all the dangerous of high affinity TcRs resulting from the risks of off-target toxicity. They address this somewhat with their specificity screens and the functional studies with the bispecific, but the study is not particularly comprehensive in screening. That's ok, but do not even address this point is quite striking.

R2_Author response 21: We thank the reviewer for this feedback and refer the reviewer to our responses to related comments 4 and 6. We have revised the manuscript with additional data (**Fig 4 and Fig 5e**) and discussion to address the concerns around cross-reactivity of high affinity TCR. For example, in the discussion section:

Line 517-573: ‘an engineered TCR directed to MAGE-A3 pHLA was associated with off-target cardiac toxicity and patient death. Subsequent analyses revealed this MAGE-A3 TCR cross reacts with a peptide derived from the muscle protein Titin⁶⁷. These examples demonstrate both the promise and the potential danger of engineered TCR therapies and highlight the importance of extensive molecular and functional understanding of TCR specificity prior to initiating clinical studies.’

R2_Comment22: Did I miss the reference in the paper to the structure with the WT TcR? Why is there no comparison of the high affinity version to this? It would seem relevant to points about specificity, etc.

R2_Author response 22: We have revised the manuscript with an expanded paragraph describing the structural comparison of the WT and high affinity TCR-HLA complexes (**lines 198-223**). To support this analysis, **supplementary figure S3** has been expanded with four new structural snapshots.

Reviewer #3:

*Poole and colleges describe and functionally characterize a novel TCR-anti-CD3 scFv fusion protein directed against a KRASG12D peptide presented by HLA-A*11. The results are well presented and the conclusions are supported by the data provided. Based on the high frequency of the KRAS-G12D mutation in many solid tumors, this study has a high translational potential. However additional experiments must be performed to demonstrate the in vivo anti-tumor function of high affinity T cell receptor targeting the KRASG12D cancer neoantigen. A disadvantage of the approach is the HLA-A*11 restriction of the ImmTAC. Even though HLA-A*11 is one of the more frequent HLA-A alleles, it is only expressed in around 5-15% of the Europeans/North Americans and 20-30% of the East Asian populations. Thus this therapy would be only be available for a minority of the patients with KRASG12D mutated tumors.*

R3_General Author Response:

We thank the reviewer for this critical feedback and positive comments on the strength of the data and high translational potential of this study. We agree with the reviewer that in western populations, the frequency of HLA-A*11:01 is lower than more prevalent HLA-A alleles, for example HLA-A*02:01. Moreover, as the reviewer notes, outside of western populations, HLA-A11:01 is more prevalent. We respectfully point out that the incidence of pancreatic and colorectal cancers and accounting for the frequency of KRAS^{G12D} mutation in those patients still means that there are a significant number of HLA-A*11:01+ patients that could derive benefit from a therapeutic agent targeting HLA-A*11:01-KRAS^{G12D}.

Specific comments:

Fig. 1A-D, Fig. 4A-G: numbers (n) of repeated experiments should be indicated, error bars are missing in Fig. 1B.

Fig. 1A: The figure legend states that the non-transduced controls are not shown, even though they appear in the graph.

R3_Author response: We thank the reviewer for bringing these errors to our attention. We have revised the manuscript and corrected the Figure legends to include numbers of repeated experiments as well as error bars in **Fig 1B**. Reference to the non-transduced cells as not shown has been removed from the figure legend.

Fig. 1B: According to the figure legend and axis description, bars in Fig. 1B represent the fold change of KD (dissociation constant) of JDI TCR towards mutated peptides relative to G12D. Consequently, when the affinity to the mutated peptide is lower than the affinity to G12D, the ratio is superior to 1. When the affinity to the mutated peptide is higher than the affinity to G12D, the ratio is inferior to 1. Most of the bars indicate values inferior to 1 and, thus, most of the mutated peptides display a higher affinity to JDI TCR than G12D. Therefore, I don't understand the author's statement in the text saying that peptide residue D6, G4, A5 and G9 are most critical for JDI TCR binding. In addition, I would have expected the affinity of G12D to JDI TCR to decrease when individual amino acids of the peptide are mutated.

R3_Author response: We thank the author to bring this to our attention. We have clarified the method used to calculate the values presented in the histogram (**Fig 1b**) both in the figure legend:

Line 1082-1083: Ratio is calculated as K_D of JDI TCR binding to KRAS^{G12D}/Alanine mutant.

The ratio is calculated as affinity (K_D) of JDI TCR to HLA-A*11-KRAS^{G12D}/Alanine mutant. Except for V1A and V3A which show slightly stronger binding to JDI TCR, all other mutants bind weakly and hence have an inferior ratio.

*Fig. 1 and Table 2: In the text Z142, the "affinity window" is defined as fold change in affinity of JDI TCR to KRASG12D-HLA-A*11 compared to KRASWT-HLA-A*11. Although, the figure legend Fig. 1D defines the "affinity window" as fold change of the affinity of JDI TCR relative to parental (non-affinity matured?) TCR. In both cases the same values of the "affinity window" are shown.*

*Table 2: The "affinity window" of JDIa9bwt in Table 2 and of the JDI TCR in Z298 was apparently calculated as KD KRASWT-HLA-A*11/ KD KRASG12D-HLA-A*11. However,*

when applying the same calculation to JDI a41b1 in Table 2 the “affinity window” should be much higher than indicated. Please indicate a clear and consistent definition of the “affinity window”.

R3_Author response: We thank the reviewer for highlighting this mistake. The affinity of JDIa41b1 TCR binding to HLA-A*11-KRAS^{G12D} was erroneously reported as 0.6E-10, where the correct value was 6E-10. This would have caused the confusion regarding the higher affinity window that the reviewer has suggested. We have modified **Table 2** and defined the affinity window throughout the manuscript.

Fig. 1/Table 2: Within the text the authors mention t1/2 measurements of affinity enhanced TCRs. Please indicate the values, f.e. in Table 2.

R3_Author response: We thank the reviewer for this suggestion and have revised the manuscript and **Table 2** to include t1/2 measurements of the TCRs listed

Fig. 2A,C,E; Fig. 3A: Superposed graphical representation of two/four (in Fig. 2E) peptides within one image are very difficult to understand. Thus, when structural differences of peptides should be highlighted and are important for the message, I would rather represent them in two images side by side than overlaid. I.e., differences are much easier to see in Fig. 3E,F where two juxtaposed images show the differences.

R3_Author Response: We thank the reviewer for this suggestion. This image shown in Fig 2e is now separated into two images (**Fig 2e and f**), showing the peptide conformations of pHLA and pHLA-TCR complexes separately.

Fig. 4: Figure legends 4A and B are interchanged

R3_Author response: We thank the reviewer for bringing this error to our attention. We have revised the manuscript, correcting the Figure legend:

Lines 1157-1162: a. Immature dendritic cells differentiated from monocytes isolated from healthy donors were transfected with mRNA encoding KRAS^{WT} or KRAS^{G12D} or **b** pulsed with indicated exogenous peptide, were treated with IMC-KRAS^{G12D} and co-cultured with autologous T cells for 24 h. T cell activation was measured by IFN γ ELISPOT assay.

Fig. 4D: Please show relative intensities of KRASG12D/GAPDH bands for different clones to allow the reader to compare expression levels with IFN γ response after co-culture with PBMCs in Fig. 4E.

R3_Author response: We thank the reviewer for this suggestion and have revised the manuscript and Fig 4E (**Fig 5C in the revised manuscript**) to include densitometry data that indicate the relative intensities of KRASG12D/GAPDH bands for the individual clones.

Fig. 4G: Were other markers than IFN γ increased such as perforin, granzyme, CD107a etc.?

R3_Author response: We thank the reviewer for this suggestion. We have revised **Supplementary Figure S9** to include detection of Granzyme B and IL-2 in addition to redirected T cell killing.

Fig 4F: The in vitro cytolytic activity is impressive but in order to understand the translational potential in vivo experiments in the mouse model using KRAS mut cancer cell lines or genetic models are required.

KRAS-G12D was recently shown to induce NLRP3 activation (Nat Commun. 2020 Apr 3;11(1):1659. doi: 10.1038/s41467-020-15497-1) – how would that affect the anti-KRAS-G12D T cell response given that continuous IL-1beta exposure may cause T cell exhaustion?

R3_Author response: The reviewer raises an interesting point that would be an interesting starting point for further studies, however the experiments are not trivial. IMC-KRAS^{G12D} is composed of two domains: a soluble TCR targeting domain and an anti-CD3 effector end. The IMC-KRAS^{G12D} soluble TCR targeting domain is highly specific for the KRAS^{G12D} peptide presented by human HLA-A*11:01, while the IMC-KRAS^{G12D} anti-CD3 effector domain is specific for human CD3. This specificity for human peptides presented as pHLA complexes and CD3 greatly limits testing in animal models, none of which recapitulate all key human features needed to assess the biologic affects of IMC-KRAS^{G12D}, This includes the production of human cytokines such as IL-1beta and their cognate human receptors. Human lymphoid tissue is also required to allow the production of a broad range of lymphoid subsets.

Comments on Methods:

*Please describe in more detail how the KRASG12D-HLA-A*11-directed TCR was identified and how cells were pulsed with peptides.*

R3_Author response: We have revised the Materials and methods section of the manuscript to provide more detail on the methodology used to identify the TCR including descriptions of how cells were pulsed with peptides. For example:

Line 578-589: ‘T cells were isolated from a HLA-A*0201/A*11+ donor and stimulated for 7 days with autologous dendritic cells that had been pulsed with 1 μM KRAS^{G12D} peptide (VVVGADGVGK) to displace naturally presented peptides. For this, the dendritic cells were incubated with exogenous peptide for two hours, then washed two times with culture media. T cells were subsequently stimulated twice more over an additional 14 day period with autologous activated B cells that had been pulsed with 0.1 μM KRAS^{G12D} peptide. T cell cultures were screened by IFNγ ELISPOT assay to identify KRAS^{G12D} peptide specific T cells. T cell clones were sorted on the basis of the expression of the activation markers CD25 and CD137 after incubation with the KRAS^{G12D} peptide by using a FACSAriaII (BD Biosciences). TCR gene sequences were identified from a specific T cell clone by rapid amplification of cDNA ends (RACE).’

Line 732-737: ‘For assays using exogenous peptide loaded cells, target cells were incubated with 10 μM peptide (unless otherwise indicated) and co-cultured with PBMC from non-HLA*A11:01/03:01 donors in the presence of IMC-KRAS^{G12D} in RPMI-1640 medium containing 25 mM HEPES and supplemented with Penicillin-Streptomycin (Gibco™) and 10% fetal bovine serum overnight at 37 °C.’

REVIEWER COMMENTS

Reviewer #1 (Remarks to the Author):

The authors have done several of the suggested improvements and clarifications. Still several statements are still an overreach and they have not been addressed .
The conclusions of the deconvoluted binding energies might be largest issues; they are not differences large enough, and the energies might not have meaning.
The following minimally would need to be addressed:

1- R1_Author response 1

The authors added a title/section as requested. In it they use the word anchor. "KRASG12D peptide D6 residue acts as a secondary HLA anchor in the HLA-A*11-KRASG12D-TCR complex". An immunologist should be consulted if 'anchor' can be used so freely.

2- R1_Author response 2, 3, 4, 6, 7, 9a, 9c, 9d, 9e, 9f, 9g, 9h, 10a, 10b, Satisfactory

3- R1_Author response 5: the authors have included a figure that still does not help. What fold does the anti CD3 scfv have? Certainly not something that looks like a transmembrane helix. Size of the icon should be commensurable to reality (so similar to the a/b tcr). Not clear what is the top doing.

Reviewer #2 (Remarks to the Author):

The authors have considerably revised the manuscript, to the point where it is almost a completely new submission. The distracting hype and spin and major experimental omissions are gone, and in most places the crucial details left out are back in. The revised manuscript allows the reader to see what is new - I commend the authors on their attention to the critiques and their efforts in revising the manuscript.

Due to the extensive changes, I have a couple of issues that remain to be addressed, including key points that are needed to support the new revised interpretations:

1) Line 161: In the description of the two ternary structures, the authors report a Ca RMSD of 1.44 Å. Is this over all atoms, including the variable and constant domains? It's actually large for a Ca measurement. This could result from structural differences in the binding interface, or simple domain reorientations that are present in the two crystal lattices. The authors should give Ca RMSD values for just the variable domains. They should also include full atom RMSDs to give us an idea of the side chain similarities, and then comment on different side chain positions - as it stands, with the 1.44 Å Ca RMSD presented, one might easily conclude there are some real differences in the interfaces that are not being shown.

2) Line 168: which side chains have weak density? Are they the same in each structure? The authors later put considerable attention on the idea of electrostatics, which can be long range, so the slight suggestion that these don't matter because they are "not in direct contact with pHLA" is a bit misleading.

3) Line 203: the conformations of the G12D peptides are "identical" in the two structures - how identical? Ca and full atom RMSDs are needed, with some clarity around any differences.

4) Lines 239-247: once again, quantify and give us some numbers around the conformational differences.

5) Lines 255-257: The statement here is a little fluffy and could use some tightening: there *will* be energetic differences, because the affinities are different, so this statement as written is a bit meaningless. One might restate it to say something along the lines of "we hypothesized that a decomposition of binding energetics could provide insight into the origin..." or something like that.

Reviewer #3 (Remarks to the Author):

The authors have addressed all my comments.

Manuscript number: NCOMMS-21-35973A; submission date 31st March 2022

Reference expertise: TCR structure/Neoantigens/structural biology/Immune therapy in KRAS driven cancer

RESPONSE TO REVIEWER COMMENTS

Reviewer #1 (Remarks to the Author):

The authors have done several of the suggested improvements and clarifications. Still several statements are still an overreach and they have not been addressed .

The conclusions of the deconvoluted binding energies might be largest issues; they are not differences large enough, and the energies might not have meaning.

The following minimally would need to be addressed:

1- R1_Author response 1

*The authors added a title/section as requested. In it they use the word anchor. “KRASG12D peptide D6 residue acts as a secondary HLA anchor in the HLA-A*11-KRASG12D-TCR complex”. An immunologist should be consulted if ‘anchor’ can be used so freely.*

R1 author response 1:

We thank the reviewer for the comment. We have now replaced the term ‘secondary anchor’ in the text:

Line 191: “KRAS^{G12D} peptide D6 side chain was buried in the HLA groove of the HLA-A*11-KRAS^{G12D} -TCR complex”

Line 196: ‘...was buried in the HLA groove’

2- R1_Author response 2, 3, 4, 6, 7, 9a, 9c, 9d, 9e, 9f, 9g, 9h, 10a, 10b, Satisfactory

R1 author response 2:

We thank the reviewer for acknowledging that our revisions have satisfactorily addressed their initial comments.

3- R1_Author response 5: the authors have included a figure that still does not help. What fold does the anti CD3 scfv have? Certainly not something that looks like a transmembrane helix. Size of the icon should be commensurable to reality (so similar to the a/b tcr). Not clear what is the top doing.

R1 author response 3:

We thank the reviewer for this suggestion. We have now modified the schematic in **Fig 1e**. to represent the anti-CD3 scFv as standard 2-domain fragment.

Reviewer #2 (Remarks to the Author):

The authors have considerably revised the manuscript, to the point where it is almost a completely new submission. The distracting hype and spin and major experimental omissions are gone, and in most places the crucial details left out are back in. The revised manuscript allows the reader to see what is new - I commend the authors on their attention to the critiques and their efforts in revising the manuscript.

R2 author response:

We thank the reviewer for the positive feedback on our efforts to improve the manuscript the manuscript by paying attention to the detailed critique in the initial review

Due to the extensive changes, I have a couple of issues that remain to be addressed, including key points that are needed to support the new revised interpretations:

1) Line 161: In the description of the two ternary structures, the authors report a Ca RMSD of 1.44 Å. Is this over all atoms, including the variable and constant domains? It's actually large for a Ca measurement. This could result from structural differences in the binding interface, or simple domain reorientations that are present in the two crystal lattices. The authors should give Ca RMSD values for just the variable domains. They should also include full atom RMSDs to give us an idea of the side chain similarities, and then comment on different side chain positions - as it stands, with the 1.44 Å Ca RMSD presented, one might easily conclude there are some real differences in the interfaces that are not being shown.

R2 author response 1:

We agree with reviewer's comment and have now removed the RMSD value of the full complexes and replaced with C α and all atom RMSD values of the TCR variable domains revising the text at **Line 160** to read:

“Root mean square deviation (RMSD) of TCR variable domains: 0.29 Å for C α atoms and 0.80 Å for all atoms”

2) Line 168: which side chains have weak density? Are they the same in each structure? The authors later put considerable attention on the idea of electrostatics, which can be long range, so the slight suggestion that these don't matter because they are "not in direct contact with pHLA" is a bit misleading.

R2 author response 2:

We agree that residues that are not in direct contact with pHLA may indeed considerably influence binding energetics through (long range) electrostatic interactions. Residues with weak density in the CDRs of JD1a41b1-HLA-A*11-KRAS^{G12D} are R28 α and Q57 α . In all cases, their side-chain positions are modeled (based on the available experimental data) in conformations that are similar to those in the JD1a41b1-HLA-A*11-KRAS^{WT} complex. Of these, only R28 α contacts the HLA directly and has a significantly different contribution to

the binding energetics (table S3). The structures indicate R28 α forms a salt-bridge with HLA E58 (see Fig S3(b)).

We agree with reviewer's opinion that our initial suggestion could be misleading and have now removed the sentence "Side chain densities for a few CDR residues that were not involved in direct contact with pHLA were relatively weak", **and revised the text beginning at Line 166:**

"For a few CDR residues (R28 α , Q57 α), the side chain densities were relatively weak (Supplementary Fig. S2), but the modeled side chain structures do not differ substantially between the JD1a41b1-HLA-A*11-KRAS^{WT} and JD1a41b1-HLA-A*11-KRAS^{G12D} complexes. Only R28 α contributes significantly to differences in binding affinity (based on the decomposition of binding energy calculated from MD simulations) and is discussed below."

Further we have revised the text to introduce a sentence in the section related to per residue binding energy decomposition:

Line 317: "R28 α , a residue with weak side-chain density in the JD1a41b1-HLA-A*11-KRAS^{G12D} complex, favours G12D binding, but this is fully compensated for by its HLA E58 salt-bridge partner (**Supplementary Table S3**). Notably, analysis of H-bonding in the MD simulations indicates that the occupancy of H-bond interactions between R28 α and E58 is low, especially in the JD1a41b1-HLA-A*11-KRAS^{G12D} complex (< 30%), consistent with the weak side-chain density observed."

3) Line 203: the conformations of the G12D peptides are "identical" in the two structures - how identical? Ca and full atom RMSDs are needed, with some clarity around any differences.

R2 author response 3:

We have now revised the manuscript to include all atom and C α RMSD values and highlighted the differences in the side chain rotamers of V3 and V8. The text beginning at **line 209** now reads:

"Superposition of the peptides gave RMSD values of 0.77 Å for all atoms and 0.23 Å for only C α atoms. Minor conformational differences were observed in the side chain rotamers adopted by peptide residues V3 and V8 in the two complexes. V3 side chain was buried in the HLA groove and V8 side chain was partially exposed to the solvent (Supplementary Fig. S4c, d)."

4) Lines 239-247: once again, quantify and give us some numbers around the conformational differences.

R2 author response 4:

We thank the reviewer for this suggestion. We have revised the text to include all atom and C α RMSD values for the peptides:

Line 253-254: “alignment of the KRAS^{WT} and KRAS^{G12D} peptides gave RMSD values of 0.66 Å for all atoms and 0.35 Å for only C α atoms”

5) *Lines 255-257: The statement here is a little fluffy and could use some tightening: there *will* be energetic differences, because the affinities are different, so this statement as written is a bit meaningless. One might restate it to say something along the lines of "we hypothesized that a decomposition of binding energetics could provide insight into the origin..." or something like that.*

R2 author response 5:

We thank the reviewer for this suggestion and have amended the sentence at **Line 263:**

“Although both peptides adopted similar conformation in TCR-bound state, we hypothesized that a decomposition of binding energetics could provide insights into the origin of the JD1a41b1 TCR’s ability to discriminate between the KRAS^{WT} and KRAS^{G12D} peptides with >4000-fold difference in affinity.”

Reviewer #3 (Remarks to the Author):

The authors have addressed all my comments.

REVIEWERS' COMMENTS

Reviewer #1 (Remarks to the Author):

The authors have successfully addressed all my comments and concerns. I commend them for their responsiveness.

Reviewer #2 (Remarks to the Author):

The authors have addressed by concerns and done a fantastic job of revising overall. I share the concern with one of the other reviewers (and I mentioned this as well in my first review) that the computational decomposition of binding free energies is a stretch and possibly of limited accuracy, but the work is appropriately framed, and citations and future work will tell us if the decomposition is of value.

Manuscript number: NCOMMS-21-35973B; submission date 19th May 2022

Reference expertise: TCR structure/Neoantigens/structural biology/Immune therapy in KRAS driven cancer

RESPONSE TO REVIEWER COMMENTS

REVIEWERS' COMMENTS

Reviewer #1 (*Remarks to the Author*):

The authors have successfully addressed all my comments and concerns. I commend them for their responsiveness.

Author response: we thank the reviewer for his comments and helpful suggestions to improve the manuscript.

Reviewer #2 (*Remarks to the Author*):

The authors have addressed by concerns and done a fantastic job of revising overall. I share the concern with one of the other reviewers (and I mentioned this as well in my first review) that the computational decomposition of binding free energies is a stretch and possibly of limited accuracy, but the work is appropriately framed, and citations and future work will tell us if the decomposition is of value.

Author response: we appreciate the reviewers kind feedback that we have addressed his concerns and done a good job. We agree that future work will advance the discussion of the deconvoluted binding energies.

Reviewer #3 (*Remarks to the Author*) (*from manuscript NCOMMS-21-35973A*):

The authors have addressed all my comments.

Author response: we are pleased to have addressed all the reviewers comments.